# Establishment of mouse model of inherited PIGO deficiency and therapeutic potential of AAV-based gene therapy

Ryoko Kuwayama[1,2], Keiichiro Suzuki [3,4,5], Jun Nakamura [3], Emi Aizawa [4], Yoshichika Yoshioka [3,6,7], Masahito Ikawa [8], Shin Nabatame [2], Ken-ichi Inoue [9], Yoshiari Shimmyo [10], Keiichi Ozono [2], Taroh Kinoshita [1,11] & Yoshiko Murakami [1✉]

Inherited glycosylphosphatidylinositol (GPI) deficiency (IGD) is caused by mutations in GPI biosynthesis genes. The mechanisms of its systemic, especially neurological, symptoms are not clarified and fundamental therapy has not been established. Here, we report establishment of mouse models of IGD caused by *PIGO* mutations as well as development of effective gene therapy. As the clinical manifestations of IGD are systemic and lifelong lasting, we treated the mice with adeno-associated virus for homology-independent knock-in as well as extra-chromosomal expression of *Pigo* cDNA. Significant amelioration of neuronal phenotypes and growth defect was achieved, opening a new avenue for curing IGDs.

[1] Yabumoto Department of Intractable disease research, Research Institute for Microbial Diseases, Osaka University, Osaka, Japan. [2] Department of Pediatrics, Osaka University Graduate School of Medicine, Osaka, Japan. [3] Graduate School of Frontier Bioscience, Osaka University, Osaka, Japan. [4] Graduate School of Engineering Science, Osaka University, Osaka, Japan. [5] Institute for Advanced Co-Creation Studies, Osaka University, Osaka, Japan. [6] Center for Information and Neural Networks, National Institute of Information and Communications Technology (NICT) and Osaka University, Osaka, Japan. [7] Center for Quantum Information and Quantum Biology, Osaka University, Osaka, Japan. [8] Department of Experimental Genome Research, Research Institute for Microbial Diseases, Osaka University, Osaka, Japan. [9] Systems Neuroscience Section, Department of Neuroscience, Primate Research Institute, Kyoto University, Kyoto, Japan. [10] Asubio Pharma Co., Ltd, Kobe, Japan. [11] Immunoglycobiology, WPI Immunology Frontier Research Center, Osaka University, Osaka, Japan. ✉email: yoshiko@biken.osaka-u.ac.jp

Over 150 kinds of glycosylphosphatidylinositol (GPI)-anchored protein (GPI-AP) are expressed on the surface of mammalian cells[1]. GPI-APs have a variety of roles, including acting as hydrolytic enzymes, adhesion molecules, receptors, protease inhibitors, and complement regulatory proteins[2]. At least 27 genes are involved in the biosynthesis and transport of GPI-APs[3], and mutations in these genes cause inherited GPI deficiency (IGD)[4–21]. IGD is a recessive hereditary disease, which mainly features neurological abnormalities. Recently, many individuals with IGD possessing *PIGO* mutations have been reported, and they show a wide range of symptoms, such as hyperphosphatasia, intellectual disability, developmental delay, epilepsy, and organ anomalies[7,22]. PIGO is a transferase that adds ethanolamine phosphate (EtNP) to the third mannose, thereby attaching GPI to proteins. Defects in PIGO thus cause decreased expression of various GPI-APs. As the complete loss of GPI biosynthesis causes early embryonic death[23,24], most individuals with IGDs exhibit partial deficiency in GPI biosynthesis. There is no animal model mimicking IGD individuals and the mechanism behind the neurological manifestations has not been clarified. In addition, there are no fundamental therapeutic interventions.

To obtain a better understanding of the pathogenic mechanism of IGD and to test potential therapeutic interventions, we here report three lines of *Pigo* knock-in (KI) mice bearing three mutations homologous to *PIGO* mutations found in IGD individuals with mild to severe symptoms[22,25,26]. The severest line of mice recapitulated the neurological phenotypes of PIGO deficiency.

Adeno-associated virus (AAV) vector-mediated gene delivery has undergone several trials and been used clinically to treat intractable monogenic diseases[27,28]. As IGD is associated with defects in normal neurological development, therapy should be initiated at a young age and the therapeutic effect should be maintained for many years. As for recombinant AAV (rAAV)-mediated gene transfer, rAAV persisting as an extra-chromosomal form in the nucleus of transduced cells is diluted through repeated cell divisions, leading to the loss of transgene expression[29,30]. Furthermore, as PIGO shares a stabilizing component, PIGF, with another transferase PIGG that adds EtNP to the second mannose, overexpression of PIGO would have a harmful effect through the decreased expression of PIGG[31]. A genome-editing strategy using the CRISPR–Cas9 system enables us to overcome these limitations of overexpression-based gene therapy. A conventional homology-directed repair (HDR)-mediated integration method is not efficient enough for effective treatment because of its poor accessibility to nondividing cells[32]. Recently, one of us developed a nonhomologous end joining (NHEJ)-based homology-independent strategy for targeted transgene integration (HITI), which allows more robust gene knock-in in both dividing and nondividing cells than for HDR[33]. However, the current version of HITI is still insufficient to cure many inherited diseases due to the low frequency of transgene insertion. To overcome this limitation and optimize the treatment for IGD, we developed a novel gene-therapy method, homology-independent targeted integration assisted with a low level of transgene expression (HITI-TE). HITI-TE is a combinational approach involving HITI-mediated *Pigo* full-coding cDNA insertion at the *Pigo* locus and *Pigo* full-coding cDNA expression driven by a weak promoter encoded in the ITR region of the AAV vector[34,35]. Using the HITI-TE method in disease model mice, we show the clear amelioration of various symptoms of IGD.

## Results

**Partial loss-of-function mutants of *Pigo* with different severities**. As clinical severity varies among those affected by IGD depending on the residual GPI biosynthetic activities of mutant alleles, we aimed to generate mouse lines exhibiting conditions with different levels of severity. To achieve this, we chose three missense mutants of the *PIGO* gene, p.Arg119Trp (R119W), p.Thr130Asn (T130N), and p.Lys1047Glu (K1047E), found in individuals with IGD having different clinical severities and generated the homologous mouse *Pigo* mutant cDNAs p.R119W p.T130N, and p.K1051E, respectively. We first analyzed the activities of the three *Pigo* mutants. *Pigo*-knockout (KO) Neuro2a cells generated by the CRISPR–Cas9 system were transfected with each mutant *Pigo* cDNA and analyzed by flow cytometry. The three missense mutant cDNAs rescued the surface expression of GPI-AP with reduced efficiency compared with the wild-type cDNA (Supplementary Fig. 1a). Protein expressions of *Pigo* mutants were comparable to the wild-type level (Supplementary Fig. 1b). Based on the percentage of GPI-AP-rescued cells, we concluded that T130N was the severest mutation, R119W and K1051E were milder. We next introduced these missense mutations into mouse embryonic stem (ES) cells by CRISPR–Cas9-mediated knock-in (Supplementary Fig. 2) and generated three *Pigo*-KI mouse lines, termed the A-, B-, and C-lines, for R119W, T130N, and K1051E, respectively (Fig. 1a–c and Supplementary Fig. 3). We also called the three mutant alleles R119W, T130N, and K1051E, alleles *a*, *b*, and *c*, respectively. During R119W KI procedures in ES cells, we obtained an ES clone having a KO allele and generated a *Pigo*-KO mouse line (Fig. 1d). By crossing each heterozygous *Pigo* KI mouse, *Pigo*$^{a/+}$, *Pigo*$^{b/+}$, and *Pigo*$^{c/+}$, with a heterozygous KO mouse, *Pigo*$^{+/-}$, we obtained knock-in/knockout (KIKO) mice, *Pigo*$^{a/-}$, *Pigo*$^{b/-}$, and *Pigo*$^{c/-}$ (Fig. 1e), which mimicked the genotype of IGD individuals with compound heterozygous bi-allelic *PIGO* mutations[22,25].

**Decreased expression of GPI-APs in *Pigo* B-line KI mice and KIKO mice**. Using the *Pigo* B-line mice with the severest mutation, we tested the expression of GPI-APs. Flow cytometric analysis of blood granulocytes indicated that the surface expression levels of Gr-1 in the homozygous *Pigo* B-line KI mice (termed *Pigo*$^{b/b}$) were decreased compared with those of their wild-type controls (79.1% of the control, $p < 0.001$). *Pigo* B-line knock-in/knockout (KIKO) mice (termed *Pigo*$^{b/-}$) showed lower levels of Gr-1 than *Pigo*$^{b/b}$ mice (66.0% of the control, $p < 0.001$) (Fig. 1f). These results were compatible with previous reports describing that patients with *PIGO* deficiency showed lower levels of GPI-APs in blood granulocytes[7,22].

To determine whether the expression of GPI-APs in the brain is reduced, the brain lysates of the *Pigo*$^{b/-}$ mouse and its wild-type littermates were analyzed for Thy1 and contactin1, GPI-APs, by western blotting. The expression of Thy1 in the *Pigo*$^{b/-}$ mouse cerebrum was decreased to about 50% of that of the control, whereas the expression of contactin1 was not significantly decreased (Supplementary Fig. 4). Additionally, we found that myelin basic protein (MBP), a major, non-GPI-anchored protein of myelin, was decreased in *Pigo*$^{b/-}$ mouse cerebrum and cerebellum to approximately 70% of control levels, suggesting that myelination is affected in the *Pigo*$^{b/-}$ mouse (Supplementary Fig. 4).

**Hyperphosphatasia seen in *PIGO*-IGD was reproduced in *Pigo* KIKO mice**. The plasma alkaline phosphatase (ALP) level was higher in homozygous *Pigo*$^{b/b}$ mice than in their wild-type controls ($p < 0.05$). *Pigo*$^{b/-}$ mice showed even more elevated levels of plasma ALP than their wild-type controls ($p < 0.001$) (Fig. 1g). As hyperphosphatasia in IGD with PIGO deficiency is caused by secretion of the processed precursor protein of ALP without GPI anchoring, its reproduction in these *Pigo*$^{b/b}$ and *Pigo*$^{b/-}$ mice indicated that GPI biosynthesis was impaired.

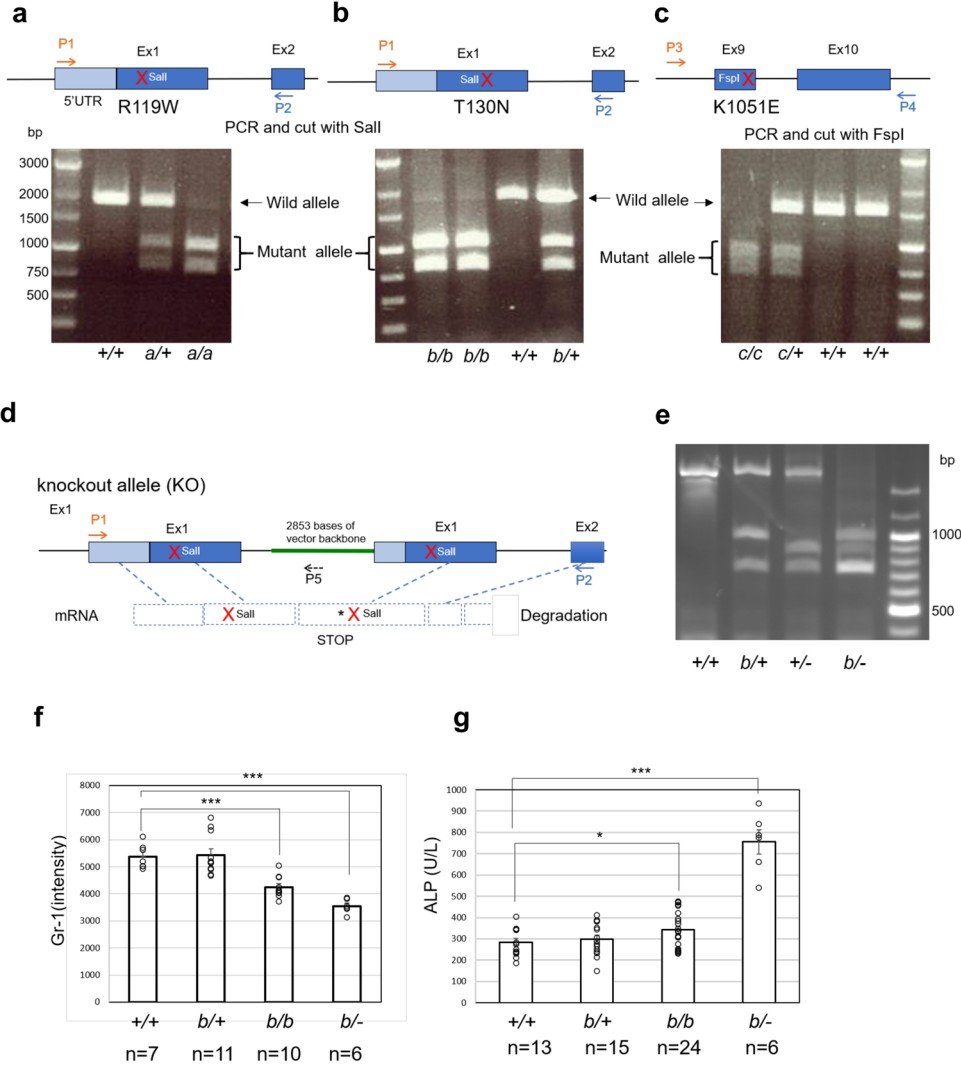

**Fig. 1 Genotyping of three lines of *Pigo* KI and KO mice and confirmation of their phenotype by flow cytometry of the granulocytes and serum ALP levels. a, b** Genotyping of *Pigo* A- and B-line KI mice by PCR with the primer 1 (P1) and the primer 2 (P2), followed by digestion with SalI. PCR products from KI alleles are sensitive to SalI showing bands of 1021 bp and 761 bp. **c** Genotyping of *Pigo* C-line KI mice by PCR with the primer 3 (P3) and the primer 4 (P 4), followed by digestion with FspI. KI alleles were detected by the FspI-digested PCR products of 831 bp and 665 bp. **a–c**, the representative results from at least three times repeated experiments. **d** Structure of *Pigo*-KO allele, which was generated due to the homologous recombination failure during A-line KI process. **e** Genotyping of *Pigo* B-line KIKO mice by PCR with the P1, P2, and the primer 5 (P5). KO alleles were detected by the SalI-digested PCR products of 878 bp and 761 bp. This is the representative result from at least three times repeated experiments. **f** Reduction in the surface expression levels of Gr-1 on blood granulocytes from *Pigo*$^{b/b}$ and *Pigo*$^{b/-}$ mice. Mean ± SEM for each group of mice is shown. *n* number of animals; ***$p < 0.001$. (*t*-test, two-sided, *P*-value, Wild:KIhomo, $6.9 \times 10^{-5}$; Wild:KIKO, $2.8 \times 10^{-6}$). **g** Hyperphosphatasia was observed in *Pigo*$^{b/b}$ and *Pigo*$^{b/-}$ mice. ALP activity was measured in plasma from mice older than 4 months. Mean ± SE for each group of mice is shown. *n* number of animals; *$p < 0.05$; ***$p < 0.001$. (*t*-test, two-sided, *P*-value, Wild:KIhomo, 0.035; Wild:KIKO, $1.1 \times 10^{-8}$) Source data are provided as a Source Data file.

**Growth defect, decreased survival, and tremor in *Pigo* KI and KIKO mice.** We followed the growth of the three lines of *Pigo* homozygous KI and KIKO mice in comparison with that of their littermate heterozygous and wild-type mice by measuring the body weight. The weight of *Pigo*$^{b/b}$ mice was similar to that of the controls at birth, but from around 4 weeks of age, their growth gradually slowed down, and their weight were significantly lower than *Pigo*$^{b/+}$ and *Pigo*$^{+/+}$ at 5–6 weeks of age ($p < 0.05$). Their growth ceased at around 7–8 weeks, and their weight gradually decreased after 14 weeks (Fig. 2a). *Pigo*$^{b/-}$ mice had stronger growth defects, and they were significantly smaller than *Pigo*$^{b/+}$ or *Pigo*$^{+/-}$ and *Pigo*$^{+/+}$ at 4 weeks of age ($p < 0.05$) (Fig. 2b). In contrast, the growth of *Pigo* homozygous KI mice of A- and C-lines (*Pigo*$^{a/a}$ and *Pigo*$^{c/c}$), and that of their KIKO (*Pigo*$^{a/-}$ and

*Pigo*$^{c/-}$) mice were comparable to the levels of their heterozygous and wild-type littermate controls (Fig. 2a, b). Life spans of *Pigo*$^{b/b}$ mice (median survival, approximately 40 weeks) and *Pigo*$^{b/-}$ mice (median survival, approximately 15 weeks) were clearly shorter than those of their wild-type controls (100% survival at 36 weeks of age), whereas most *Pigo*$^{a/a}$, *Pigo*$^{a/-}$, *Pigo*$^{c/c}$, and *Pigo*$^{c/-}$ survived for more than 36 weeks, which was not so different from the findings of their littermate controls (Fig. 2c).

Tremor occurred in *Pigo*$^{b/b}$ mice without exception at around 6 weeks, while it started a week earlier (at 5 weeks) in *Pigo*$^{b/-}$ mice (Fig. 2d and Supplementary Movie 1 and 2). In A- and C-lines, homozygous KI mice never showed tremor, but almost all *Pigo*$^{a/-}$ mice started tremor at 7–16 weeks of age. *Pigo*$^{c/-}$ mice did not exhibit tremor (Fig. 2d). All of the *Pigo*$^{b/b}$ and *Pigo*$^{b/-}$

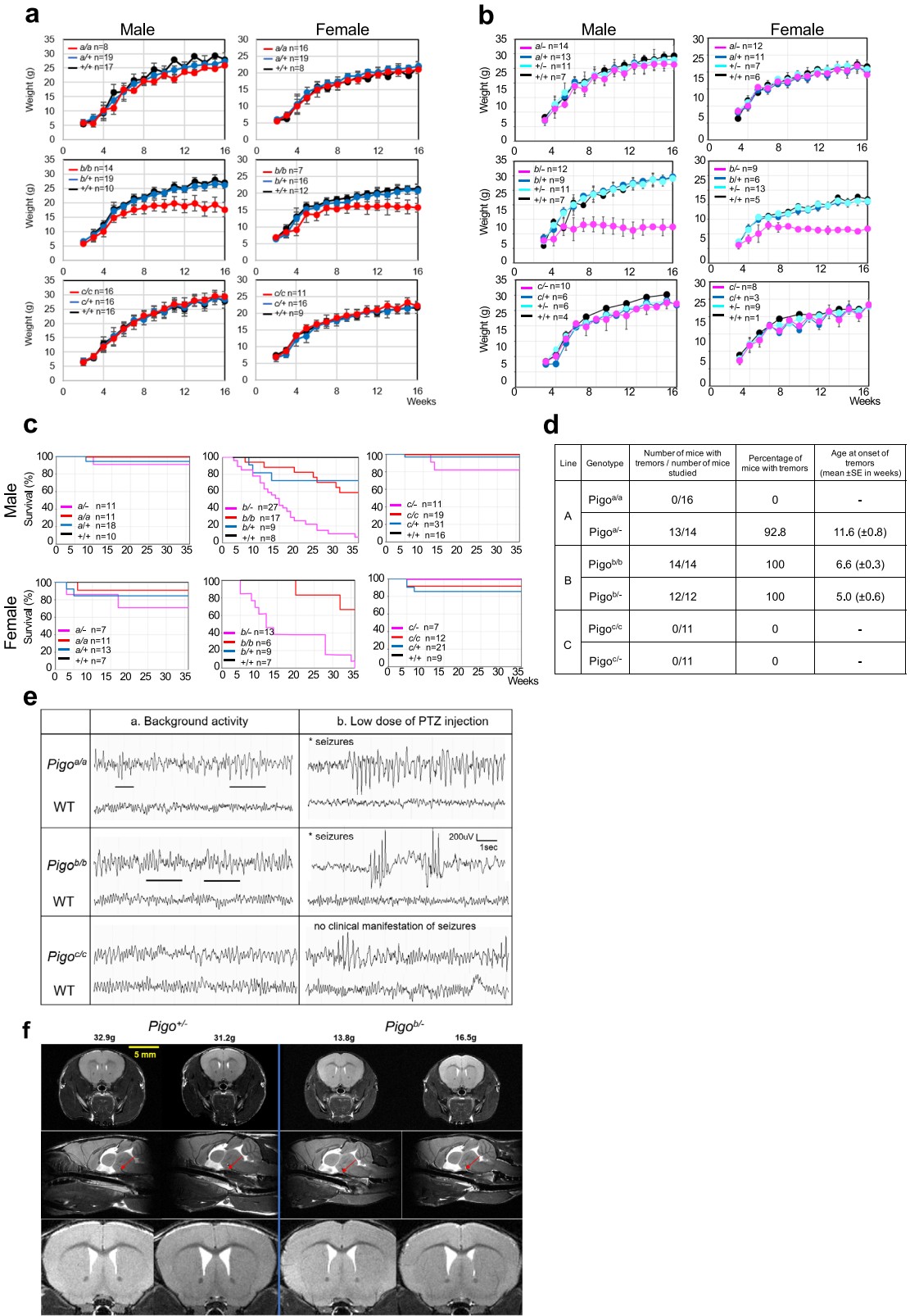

mice displayed hindlimb clasping at around 12 weeks of age (Fig. 4e) and they started to exhibit an ataxic gate with dragging of the hindlegs (Supplementary Movie 3).

**The *Pigo* KI mice were more susceptible to seizures**. We analyzed the 24-h EEG record with video of *Pigo* KI mice and wild-type controls (n = 3, each group) (Fig. 2e). *Pigo^{a/a}* and *Pigo^{b/b}* mice exhibited slow waves in the 3–4-Hz range compared with the 6–7-Hz activity in wild-type controls, whereas two out of three *Pigo^{c/c}* mice showed those in the 3–4-Hz range. Fast Fourier transform (FFT) power spectral analysis of 8-h recorded EEG in the dark phase showed that all three lines of *Pigo* KI mice showed high power in delta waves (0.75–4 Hz), indicating that slow waves

**Fig. 2 Phenotypes of *Pigo* KI and KIKO mice. a** Comparison of weekly body weight in three KI groups respectively (*Pigo* A-, B-, and C-line). Data are presented as mean values + SD. **b** Comparison of weekly body weight in three KIKO groups respectively (*Pigo* A-, B-, and C-line). Data are presented as mean values + SD. **c** Kaplan–Meier survival curves for *Pigo* KI and KIKO mice of A-line (left), B-line (center) and C-line (right). *n* number of animals. **d** Comparison of tremors in three KI and KIKO groups respectively (*Pigo* A-, B-, and C-line). **e** Representative EEG recordings in *Pigo* homozygous KI mice. Images showing the background activity in *Pigo* KI mice and their wild-type controls (each, $n = 3$). Horizontal bars, interictal epileptiform discharges were found in $Pigo^{a/a}$ and $Pigo^{b/b}$ mice. Seizure susceptibility induced by single injection of a low dose of PTZ (20 mg/kg) (each, $n = 3$). **f** In vivo $T_2$ weighted brain MRI of $Pigo^{b/-}$ mice compared to their heterozygous KO littermates (male, $Pigo^{b/-}$, $n = 2$; $Pigo^{+/-}$, $n = 2$). Pituitary glands (red arrows) were small in $Pigo^{b/-}$ mice (enlarged images are shown in Supplementary Fig. 6d). Source data are provided as a Source Data file.

with high amplitude were dominant throughout the dark phase. In wild-type mice, such peaks in slow waves could not be seen (Supplementary Fig. 5a). $Pigo^{a/a}$ and $Pigo^{b/b}$ mice displayed interictal epileptiform discharges such as spikes and sharp waves (Fig. 2e). These were consistent with the slow background activities with sharp waves that were previously observed in individuals with PIGO IGD[22,36]. During video observation, myoclonic seizures with epileptic discharges were displayed in $Pigo^{a/a}$ and $Pigo^{b/b}$ mice, and clonic seizures, a more severe type, were seen only in $Pigo^{b/b}$ mice. No spontaneous seizures were seen in $Pigo^{c/c}$ mice nor in wild-type mice. To assess the susceptibility of the mutant lines to seizures, we confirmed the presence of acute PTZ-induced seizures using scoring new PTZ scale (Supplementary Table 1) by referring to Modified Racine score[37]. After the administration of low doses of PTZ (20 mg/kg), all $Pigo^{b/b}$ mice displayed severe myoclonic jerk or clonic seizures (score 3–6) accompanied by electrographic activity such as spike and wave complex (Fig. 2e). In contrast, one out of three $Pigo^{a/a}$ mice showed myoclonic jerk (score 2–3). $Pigo^{c/c}$ mice showed only epileptic discharges in EEG, without behavioral manifestation of seizures (score 1) (Fig. 2e).

**Magnetic resonance imaging (MRI) analysis of mouse brain.** Brains of $Pigo^{b/-}$ mice and littermate $Pigo^{b/+}$ mice were analyzed by MRI. No mice showed any difference in the contrast in $T_2$-weighted images (Fig. 2f). The corpus callosum and anterior commissure were thinner in $Pigo^{b/-}$ male mice than in the controls (Fig. 2f). $Pigo^{b/-}$ mice showed a decreased volume of skeletal muscle in the head (82% of control) (Supplementary Fig. 6a). It appears that the decreased muscle volume accounted for the decreased body weight in $Pigo^{b/-}$ mice (Supplementary Fig. 6b). The volumes of the cerebellum were slightly decreased in $Pigo^{b/-}$ mice (90% of control), whereas those of the cerebrum were not (Fig. 2f and Supplementary Fig. 6c). In addition, the pituitary glands were small in $Pigo^{b/-}$ mice (Fig. 2f and Supplementary Fig. 6d).

**Histological analysis of the brain.** To analyze the more details of the brain tissue, Nissle and immunohistochmical staining of brain sections were performed. Nissle staining, which stains the neuronal cells, showed no abnormalities in the $Pigo^{b/-}$ mouse including Purkinje cells in the cerebellum (Supplementary Fig. 7a). Immunohistochemical staining of the brain sections of $Pigo^{b/-}$ mouse showed no abnormalities including the number of oligodendroglias, astrocytes and synapses (Supplementary Fig. 7b–d). We did not find clear difference in MBP staining (Supplementary Fig. 7b) in spite that MBP of the $Pigo^{b/-}$ mouse brain was decreased to 70% of wild-type mice by western blotting (Supplementary Fig. 4). A small decrease might not be detected by immunohistochemical staining. Primary culture of neurons from $Pigo^{b/b}$ fetal brain revealed that the cells had defect in attachment to the dish, however, after attachment, they developed normally (Supplementary Fig. 8a, b). FACS analysis of the established embryonic fibroblasts from $Pigo^{b/-}$ mouse also showed

that GPI-AP stained by FLAER was not decreased compared to the controls (Supplementary Fig. 8c).

**Animal behavioral analysis.** *Pigo* B-line $Pigo^{b/b}$ and $Pigo^{b/-}$ mice were small compared to wild-type controls. They had severe tremors and apparent motor dysfunction. So, we thought that they are not suitable for comprehensive behavioral tests and chose milder *Pigo* A-line mice for behavioral tests. Their body weight was similar among the different genotypes (Supplementary Fig. 9a). $Pigo^{a/a}$ mice had no apparent muscle weakness, whereas $Pigo^{a/-}$ mice showed severe defects in motor function such as balance and coordination in the rotarod test and the four-limb hanging test (Supplementary Fig. 9b–d). In the novel object recognition test, only male wild-type mice spent more time exploring the novel objects than the familiar ones, which means that their memory was functioning normally, whereas $Pigo^{a/a}$ and $Pigo^{a/-}$ mice showed poor cognitive function (Supplementary Fig. 9e).

**Development of novel gene therapy method for IGD.** As the above results indicated that $Pigo^{b/b}$ and $Pigo^{b/-}$ mice had clearly reduced GPI biosynthesis and phenocopied symptoms seen in IGD individuals, we next investigated whether in vivo rescue of GPI biosynthesis ameliorated any of these symptoms. To achieve this, we devised the current HITI method for targeted insertion of *Pigo* cDNA including the full-coding region and 3′ UTRs (a donor sequence) into the 5′ UTR of mutant *Pigo* genes in order to rescue the KI or KO mutations (Fig. 3a). If the targeted insertion occurs as expected, *Pigo* cDNA would be transcribed under the control of the endogenous *Pigo* promoter and the Pigo protein would be generated. A donor/gRNA plasmid for the HITI method, termed pAAV-mPigoHITIgRNA, contained a donor sequence sandwiched by inverted gRNA target sequences, and a guide RNA cassette (Fig. 3a). Cas9 expression plasmid (pAAV-nEFCas9) and donor/gRNA plasmid (pAAV-mPigoHITIgRNA) were packaged into AAV-PHP.eB and generated AAV-PHP.eB-nEFCas9 and AAV-PHP.eB-mPigoHITIgRNA viruses, respectively. AAV-PHP.eB is the capsid variant of AAV9, which can efficiently transduce the central nervous system via systemic administration[38].

In addition to Pigo expression from the inserted *Pigo* cDNA (Fig. 3a, bottom), we expected Pigo expression from extra-chromosomally maintained donor /gRNA AAV (AAV-PHP.eB-mPigoHITIgRNA) because the donor construct included the full-coding *Pigo* cDNA and it would be driven by weak promoter activity in the ITR region of the AAV vector[35] (Fig. 3a, top). To test this possibility, pAAV-mPigoHITIgRNA was transfected into the *PIGO*-KO HEK293 cells by lipofection, followed by flow cytometry of CD59, a GPI-AP. The donor plasmid (pAAV-mPigoHITIgRNA) alone restored the surface expression of CD59 in *PIGO*-KO HEK293 cells, although the expression level was lower than that of a positive control, pMEpuro-mPigo, SRα promoter-driven *Pigo* cDNA (Fig. 3b). Infection of the donor AAV alone also restored the surface expression of CD59 in *PIGO*-

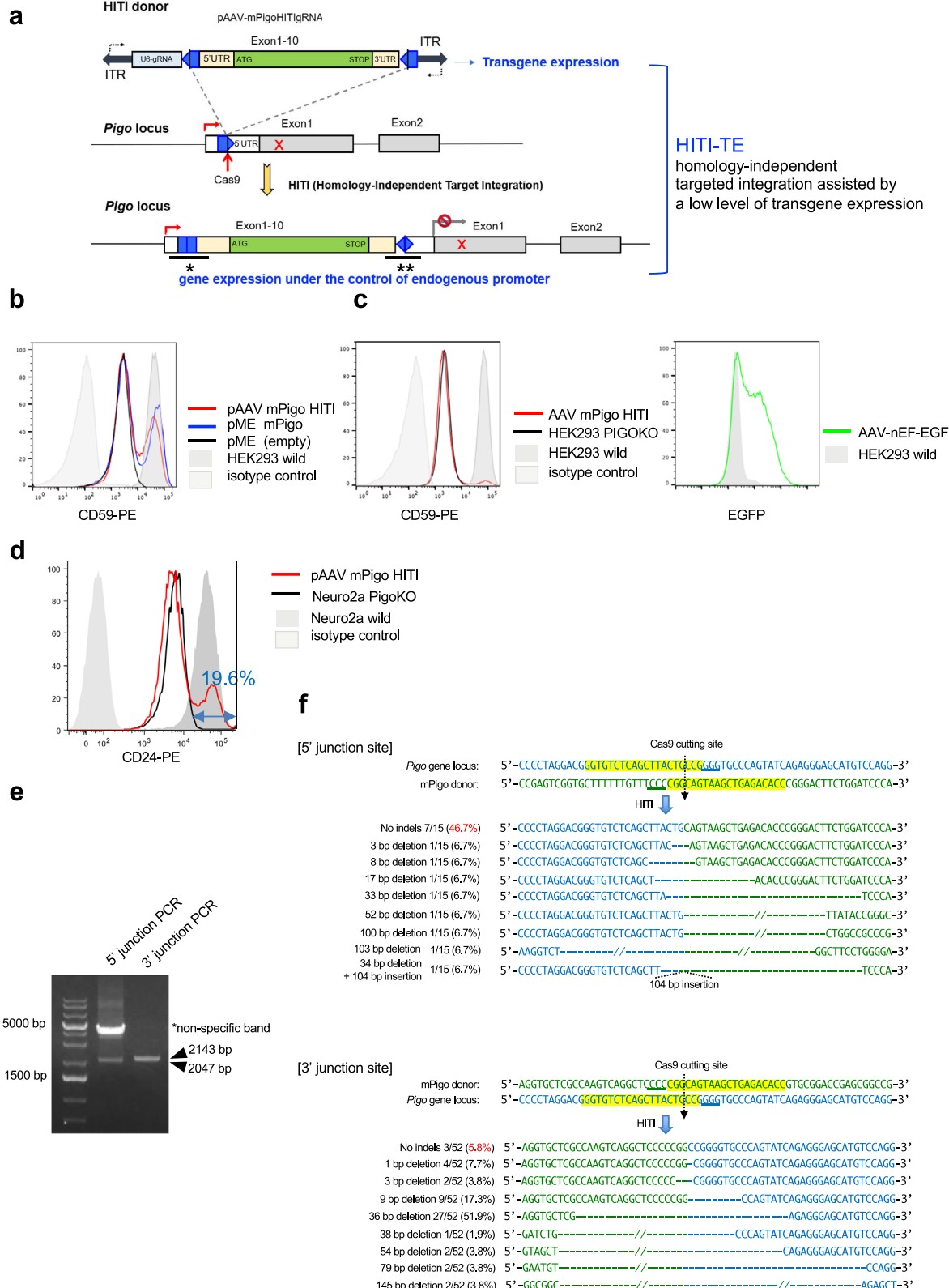

KO HEK293 cells, suggesting the weak ITR-driven expression of the full-length *Pigo* cDNA (Fig. 3c). We next compared promoter activities of AAV ITR and the endogenous *Pigo*. NIH3T3 cells were infected at various MOI with AAV-PHP.eB bearing nEF promoter-driven EGFP to measure the infection efficiencies by flow cytometry (Supplementary Fig. 10a, b). In parallel, NIH3T3 cells were infected at various MOI with donor AAV. The *Pigo*

mRNA levels were determined by RT-qPCR (Supplementary Fig. 10c) and *Pigo* gene copy numbers in the infected cells were also determined by qPCR (Supplementary Fig. 10d). When the cells were infected with MOI $10^3$, *Pigo* expression was comparable to endogenous level with 14.4% of cells infected. Increase of *Pigo* expression and *Pigo* gene copy number from non-infected cells corresponds the viral derived expression and gene copy number,

**Fig. 3 Effect of HITI-TE method in vitro. a** Schematic view of HITI-TE. (Top) The donor vector, pAAV-mPigoHITIgRNA, containing U6 promoter-driven gRNA (light blue box) and *Pigo* cDNA (light beige boxes, 5′ and 3′ UTRs; green box, coding region) sandwiched by inverted gRNA target sequences (blue pentagons); thick arrows, vector arms. (Middle) A part of mutant *Pigo* gene. White and gray boxes, 5′UTR and coding region in exons, respectively; thin lines, upstream regulatory region and introns; red X, a mutation; blue pentagon, the gRNA target sequence; red straight arrow, Cas9 cleavage site. (Bottom) The targeted mutant *Pigo* gene. Cas9 induced double strand breaks in the target site within the mutant gene and the inverted target sites in the donor vector. After repair, the functional *Pigo* cDNA was inserted into the target site in a Cas9/gRNA-resistant manner. **b** Restoration of GPI-anchored protein expression after transfection with pAAV-mPigoHITIgRNA (pAAV-mPigoHITI) donor into *PIGO*-KO HEK293 cells. Two days later, CD59 expression was analyzed by flow cytometry. Representative data from three independent experiments. **c** Left panel, restoration of GPI-anchored protein expression after infection with AAV-PHP.eB-mPigoHITIgRNA into *PIGO*-KO HEK293 cells. Two days later, CD59 expression was analyzed by flow cytometry. Right panel, EGFP expression showing transduction efficiency. Representative data from two independent experiments. **d** FACS analysis of the restored expression of CD24, a GPI-AP, on the *Pigo*- knockout Neuro2a by gene editing. *Pigo*-knockout Neuro2a were transfected with pAAV-nEFCas9 (pAAVCas9) and pAAV-mPigoHITIgRNA by lipofection. Two days later, cells were analyzed by FACS. **e** Validation of correct genome editing by genomic PCR in the in the Neuro2a cells shown in (**d**). The scheme of PCR is shown in Supplementary Fig. 12a. Asterisk shows a non-specific band. This result is a representative data of at least three times repeated experiments. **f** Sequencing analysis of 5′ junction (* in **a**) and 3′junction (** in **a**) of the integration sites in the transfected Neuro2a cells.

respectively. Virus derived *Pigo* mRNA level per viral gene was about 1% ~ 5% of endogenous level (0.82/0.24 vs 1/0.003 at MOI $10^3$), suggesting that many virus don't express *Pigo* or ITR promoter activity is much lower in NIH3T3 cells.

We next compared the AAV ITR-driven *Pigo* expression with the endogenous *Pigo* expression in the mouse brain. $1.0 \times 10^{11}$ genome copies (GCs) of AAV donor was administered via intravenous injection into newborn wild-type mice. Two weeks after injection with AAV donor, treated mice and non-treated control mice were sacrificed and whole brains were taken. RNA and genomic DNA were isolated from the cerebrum and the relative copy number of ITR-driven *Pigo* mRNA per gene was compared with that of endogenous *Pigo* mRNA per gene (Supplementary Fig. 11a–e). Relative ITR-driven expression per gene was similar to endogenous *Pigo* expression per gene, suggesting that the viral derived *Pigo* expression was comparable to the endogenous level in the cerebrum (Supplementary Fig. 11c, e).

These lines of evidence suggest that wild-type *Pigo* would be expressed from both on-target integrated *Pigo* cDNA and the extra-chromosomally maintained donor AAV, and we termed this combinational approach homology-independent targeted integration assisted by a low level of transgene expression (HITI-TE) (Fig. 3a).

To confirm that the HITI-TE system works properly, pAAV-mPigoHITIgRNA was co-transfected with a Cas9 expression plasmid (pAAV-nEFCas9) into *Pigo*-KO Neuro2a cells by lipofection. The expression of CD24, a GPI-AP, was restored in 19.6% of cells (Fig. 3d). PCR and Sanger sequencing results showed that the transgene was correctly inserted at the target sites within the 5′UTR, in association with extra small indels (Fig. 3e, f).

**Therapeutic ability of HITI-TE for *Pigo*$^{b/b}$ and *Pigo*$^{b/-}$ mice.** A single dose of $1.0 \times 10^{11}$ genome copies (GCs) of each of AAV-PHP.eB-nEFCas9 and AAV-PHP.eB-mPigoHITIgRNA was administered via intravenous injection into newborn *Pigo*$^{b/b}$ and *Pigo*$^{b/-}$ mice. At 4 months of age, the levels of Gr-1 on blood granulocytes from HITI-TE-treated *Pigo*$^{b/b}$ and *Pigo*$^{b/-}$ mice increased by 22 and 21% compared with those on granulocytes from untreated *Pigo*$^{b/b}$ and *Pigo*$^{b/-}$ mice, respectively (Fig. 4a). The plasma ALP level was significantly decreased in HITI-TE-treated *Pigo*$^{b/-}$ mice at 4 months of age (Fig. 4b, right, $p < 0.01$). It also tended to be decreased in the treated *Pigo*$^{b/b}$ mice (Fig. 4b, left). These results suggested that HITI-TE treatment increased GPI biosynthesis.

The body weight of untreated and treated mice was tracked weekly until 24 weeks of age or until death (Fig. 4c). HITI-TE treatment rescued the growth defect in both male and female *Pigo*$^{b/b}$ or *Pigo*$^{b/-}$ mice. Especially in males, the HITI-TE-treated *Pigo*$^{b/b}$ mice were significantly heavier than the untreated mice and were even similar to wild-type mice until 24 weeks of age (Fig. 4c, d).

All untreated *Pigo*$^{b/b}$ mice exhibited hindlimb clasping, whereas none of HITI-TE-treated *Pigo*$^{b/b}$ mice showed hindlimb clasping at four months of age (Fig. 4e). Tremor was significantly milder in treated *Pigo*$^{b/b}$ mice than in untreated *Pigo*$^{b/b}$ mice (Fig. 4f). As for the FFP power spectral analysis of EEG, three out of four HITI-TE-treated mice showed the decrease in slow waves with high amplitude (Supplementary Fig. 5b). Apparent spontaneous seizures were not detected during observation and only nonmotor seizures were induced by a low dose (20 mg/kg) of PTZ (score 0–3), suggesting that HITI-TE treatment was effective at reducing the susceptibility of *Pigo*$^{b/b}$ mice to seizures. The results of the hanging test in HITI-TE-treated *Pigo*$^{b/b}$ mice showed clear improvements, suggesting that the low muscle strength and/or poor motor coordination were ameliorated (Fig. 4g and Supplementary Movie 3). These results indicate that most of the phenotypes of *Pigo*$^{b/b}$ mice are reversible when *Pigo* activity is restored soon after birth, and that the novel HITI-TE therapy is a promising approach for the curative therapy of IGD.

**Evaluation of HITI-based knock-in event in vivo.** To detect the correctly targeted insertion of ectopic *Pigo* cDNA (Fig. 5a), genomic DNA was extracted from the brain and liver of HITI-TE-treated and untreated *Pigo*$^{b/b}$ mice. Genomic PCR to specifically amplify the 5′ and 3′ junctions generated the expected PCR products only from the HITI-TE-treated mouse organs (Fig. 5b and Supplementary Fig. 12b). Sanger sequencing analyses of the PCR products revealed that HITI-mediated targeted gene knock-in occurred in the liver and brain even though the indel efficiency at the 5′ and 3′ junctions varied (Fig. 5c, d and Supplementary Fig. 12c, d). Notably, the correct knock-in event was observed by PCR in the brain (Fig. 5b–d), which suggests that gene correction in the brain can be achieved through intravenous injection using the AAV-PHP.eB-assisted HITI system.

Next, we examined the expression ratio of wild-type *Pigo* transcript as well as the potential off-target effect of donor DNA and transcription from the extra-chromosomally maintained donor DNA. We used 5′ RACE and sequencing to identify the upstream sequence of exon 1 of *Pigo* mRNA transcribed from the integrated and nonintegrated donor DNAs in the liver and brain (Fig. 6a). Approximately 7–10% of transcribed products contained a normal exon 1 sequence, indicating that they were derived from the AAV donor (AAV-mPigoHITIgRNA) and were expected to show therapeutic effects (orange and red sections in Fig. 6b). Among them, 60–80% and 30% of the products from the brain and liver mRNAs, respectively, were transcribed from the

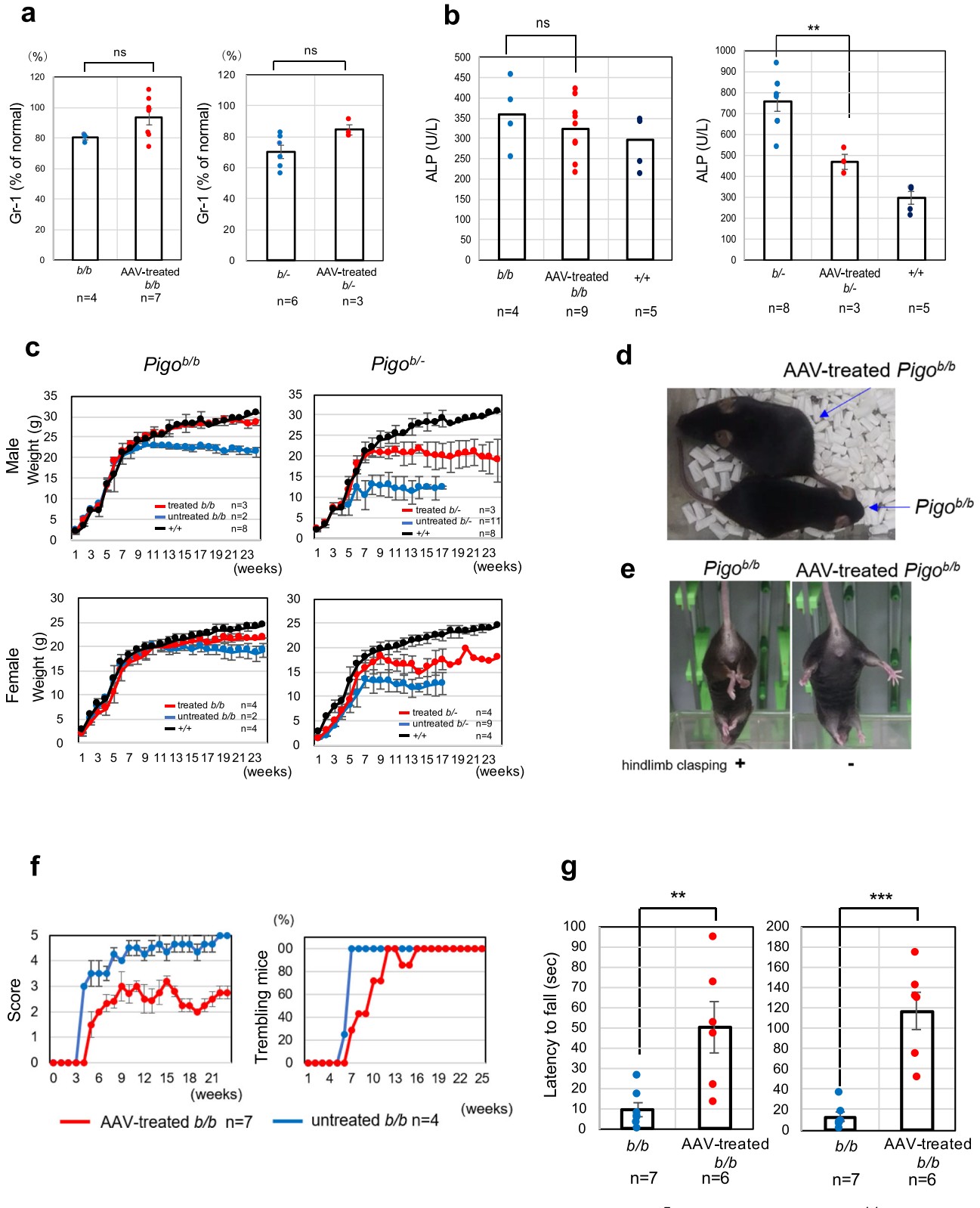

extra-chromosomal AAV donor. The possible products from the correctly integrated *Pigo* cDNA accounted for maximally 2–5% of the total products (red). The products from off-target constructs and/or of trans-splicing events were found only in the sample from the liver (gray), as also shown in a previous report[39]. Transcripts from Neuro2a cells transfected with Cas9 and AAV donor in Fig. 3d were similarly analyzed and 10% of the total products were the correctly integrated *Pigo* cDNA with no products of off-target constructs (Fig. 6b). These lines of evidence suggested that products from both on-target integrated and extra-chromosomal *Pigo* cDNAs contributed to the ameliorated phenotypes of the HITI-TE-treated *Pigo*^b/b mice.

**Fig. 4 Improvement of GPI-AP biosynthesis and the phenotypes in $Pigo^{b/b}$ and $Pigo^{b/-}$ mice after gene therapy. a** Elevation of Gr-1, a GPI-AP, levels on blood granulocytes from $Pigo^{b/b}$ mice at 4 months after AAV treatment compared to untreated mice. $n$ number of animals; ns not significant. Data are presented as mean values + SEM. ($t$-test, two-sided). **b** Amelioration of hyperphosphatasia. (**$p < 0.01$). Data are presented as mean values + SEM. ($t$-test, two-sided, $P$-value, $5.5 \times 10^{-3}$). **c** Improved growth in both treated $Pigo^{b/b}$ mice and $Pigo^{b/-}$ mice. $P$-values of $Pigo^{b/b}$: $Pigo^{b/b}$+HITI in males, $p < 0.05$ in 9–12w, $p < 0.01$ in 13–18w. $P$-value of $Pigo^{b/-}$: $Pigo^{b/-}$+ HITI in males, $p < 0.05$ in 5–16w, $p < 0.01$ in 8w and 13w, $p < 0.001$ in 7w; In females, not statistically significant in both genotypes between HITI treated and untreated. Data are presented as mean values + SD. **d** Visual comparison of HITI-TE-treated and untreated $Pigo^{b/b}$ mouse. **e** Comparison of hindlimb clasping between HITI-TE-treated and untreated $Pigo^{b/b}$ mice. **f** Severity of tremors in HITI-TE-treated and untreated $Pigo^{b/b}$ mice. Score 1–5; score 1, the mildest; score 5, the severest. Data are presented as mean values + SD. **g** Comparison of the latency to fall by the hanging test in HITI-TE-treated and untreated $Pigo^{b/b}$ mice. **$p < 0.01$; ***$p < 0.001$ ($t$-test, two-sided, $P$-value, KIhomo:KIhomo+AAV, $6.3 \times 10^{-3}$; KIKO:KIKO + AAV, $1.0 \times 10^{-4}$) Data are presented as mean values + SEM. Source data are provided as a Source Data file.

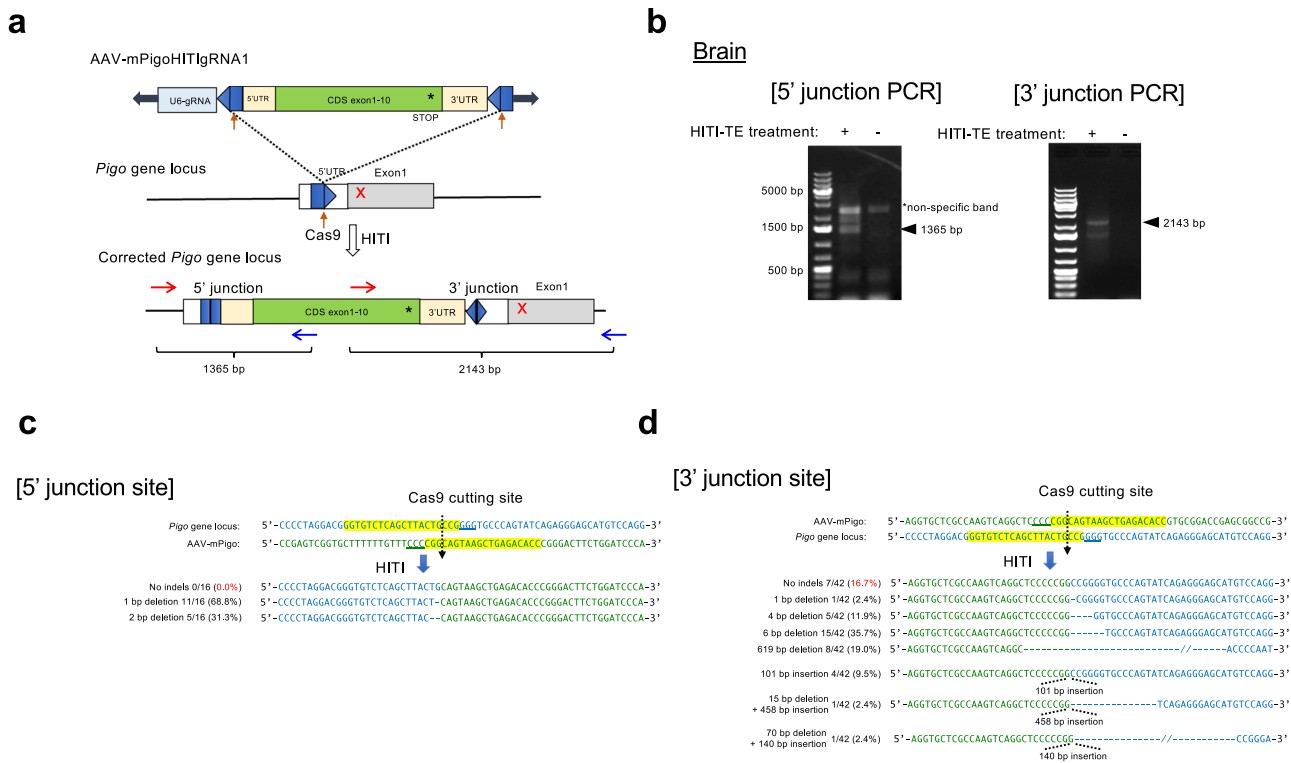

**Fig. 5 HITI-mediated genome editing via systemic injection in $Pigo$b/b mice. a** Schematic representation of the $Pigo$ KI allele b (T130N, c.389 C > A) genome editing by a HITI donor. After genome editing mediated by NHEJ-mediated HITI, the $Pigo$ full-coding cDNA including corrected mutation are inserted in 5'UTR, upstream of mutated exon 1. Blue pentagon, Pigo exon 1 gRNA target sequence in $Pigo$ 5'UTR. Red arrow or black line within blue pentagon, Cas9 cleavage site. Red and blue arrows, PCR primers. **b** Validation of correct genome editing by genomic PCR in the brain (27-week-old mice). Asterisk shows non-specific band. This result is a representative data of at least three times repeated experiments. **c** Sequencing analyses of 5' junctions of the integration sites in the brain of HITI-TE-treated $Pigo^{b/b}$ mice. **d** Sequencing analyses of 3' junctions of the integration sites in the liver and brain of HITI-TE-treated $Pigo^{b/b}$ mice.

**Possible advantage of HITI-TE in gene therapy for IGD.** To further test usefulness of HITI-TE in gene therapy of IGD, we compared the efficacies of HITI-TE and single administration of AAV-donor alone in vitro and in vivo experiments. $Pigo$-KO Neuro2a cells were transfected with pAAVCas9 and pAAV donor (pAAV-mPigoHITIgRNA) or pAAV donor only, and analyzed by FACS to determine GPI-AP-restored population for 18 days (Fig. 7a). The rescue effect of AAV donor only was clearly seen on day 3 and disappeared within 2 weeks most likely due to the dilution of pAAV donor plasmid by cell proliferation. In contrast, the cells transfected with both pAAVCas9 and pAAV donor stably retained the restored population (20% of total) for 18 days. These restored population had the corrected $Pigo$ gene by HITI system as shown in Fig. 6b. This result strongly suggests the advantage of HITI-mediated integration over ITR-mediated ectopic expression for dividing cells in terms of therapeutic duration of rescued GPI-AP expression.

To test the therapeutic contribution of ITR-dependent $Pigo$ expression in mice, $Pigo^{b/-}$ mice were treated with AAVCas9 and AAV donor (i.e. HITI-TE treated) or treated with AAV donor alone (i.e., ITR-dependent $Pigo$ expression), and their body weights were monitored. In the first 8 to 10 weeks, there were similar ameliorating effects in both groups, however, the body weight of mice treated with AAV donor only tended to decline from 10 ~ 12 weeks after the treatment (Fig. 7b). Restoration of Gr-1 expression on the granulocytes was significant in HITI-TE treated mice but not in mice treated with AAV donor only (Fig. 7c). Both groups of mice showed significant decrease in plasma ALP level in that a slightly bigger decrease in the mean ALP level was obtained by HITI-TE treatment than by AAV donor only treatment (38% vs 30%) (Fig. 7d). AAV donor only treated $Pigo^{b/b}$ mice showed clearly improved performance in the hanging test with both 5 mm and 11 mm meshes (Fig. 7e) compared with the untreated $Pigo^{b/b}$ mice (data in Fig. 4g). In

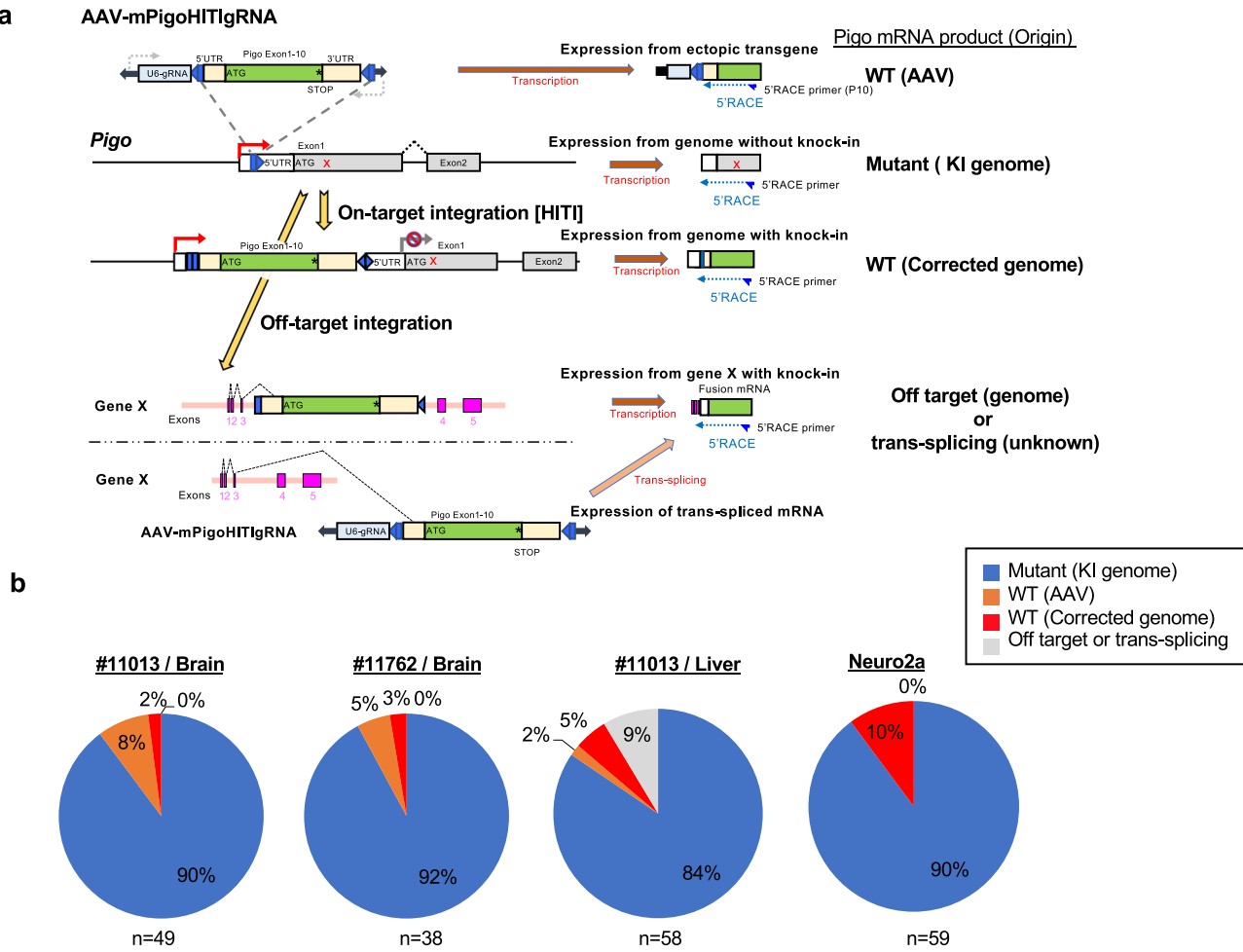

**Fig. 6 5′RACE analysis in HITI-TE-treated *Pigo^{b/b}* mice and HITI-TE transfected Neuro2a cells. a** Schematics of four or five different origins of *Pigo* transcripts generated after systemic injection of AAVs for HITI-TE-mediated gene-correction strategy. AAV-mPigoHITIgRNA includes a full-coding region of *Pigo* cDNA and potentially expresses a wild-type *Pigo* mRNA via a promoter activity of ITR. The on-target integration of the donor via HITI leads to expression of wild-type *Pigo* mRNA. The off-target integration of the donor captures the transcript of the integration site and express as a fusion gene. Alternatively, trans-splicing between off-target gene and *Pigo* cDNA also produce a fusion gene. The captured exons including from AAV-derived *Pigo*, exogenous *Pigo* and unknown off-target gene were determined with 5′RACE and sequencing. Blue half-arrows, PCR primers for 5′RACE. **b** Percentages of various *Pigo* transcripts in brain and liver RNAs from HITI-TE-treated *Pigo^{b/b}* mice (#11013; 28 weeks of age, #11762; 27 weeks of age) and HITI-transfected Neuro2a cells in Fig. 3d. Origins of the transcript species: blue, mutant *Pigo* allele *b*; orange, wild-type *Pigo* from the extra-chromosomal AAV donor; red, wild-type *Pigo* from the corrected genomic *Pigo* locus; gray, off-target *Pigo* transcripts or trans-splicing products. Percentages of respective transcript species are indicated. The numbers of sequenced 5′RACE products (*n*) are indicated below the charts. Source data are provided as a Source Data file.

both tests, HITI-TE treated mice had slightly longer mean hanging time than AAV donor only treated mice (139 s vs 126 s with 5 mm mesh and 180 s vs 134 s with 11 mm mesh) (Fig. 7e). Four out of six mice treated with AAV donor hung unstably during the tests, whereas all the HITI-TE treated mice hung stably and moved around. HITI-TE treatment ameliorated tremor better than AAV donor only treatment in both timing of tremor appearance (HITI-TE, day8 vs donor only, day6) and the severity scores (Fig. 7f and Fig. 4f). These results indicate that both HITI-mediated integration and ITR-driven transient expression of *Pigo* from the AAV donor contributed to the therapeutic effects with HITI-TE treatment.

## Discussion

Here, we describe PIGO-deficient mouse lines harboring three different human mutations. In human cells, Arg119Trp mutant showed the most severe decrease in activity leading to the severest

clinical symptoms, such as intractable seizures, severe intellectual disability, and multiple organ anomalies[22], whereas, in the murine cells, the severest was the Thr130Asn mutant (Supplementary Fig. 1). PIGO makes complex with PIGF, which is critical for PIGO activity. A possible explanation for the difference is that these mutations affect the conformation of PIGO/PIGF complex differently in human and mouse systems. Although the mouse model showed no organ anomalies nor facial dysmorphic features, the neurological symptoms recapitulated the characteristic features of *PIGO* deficiency. Additionally, brain MRI showed thin corpus callosum and anterior commissure, which mimicked the IGD phenotype. They also showed small pituitary glands, which were previously described in a PIGQ-deficient patient[40]. Ataxic gate and tremor appeared depending on the severity, which suggested cerebellar defects. A known cause of tremor is the loss of Purkinje cells[41]. Defective Purkinje cell development was seen in recently reported conditional *Piga* knockout model mice[42]. However, we did not find histological abnormalities of the

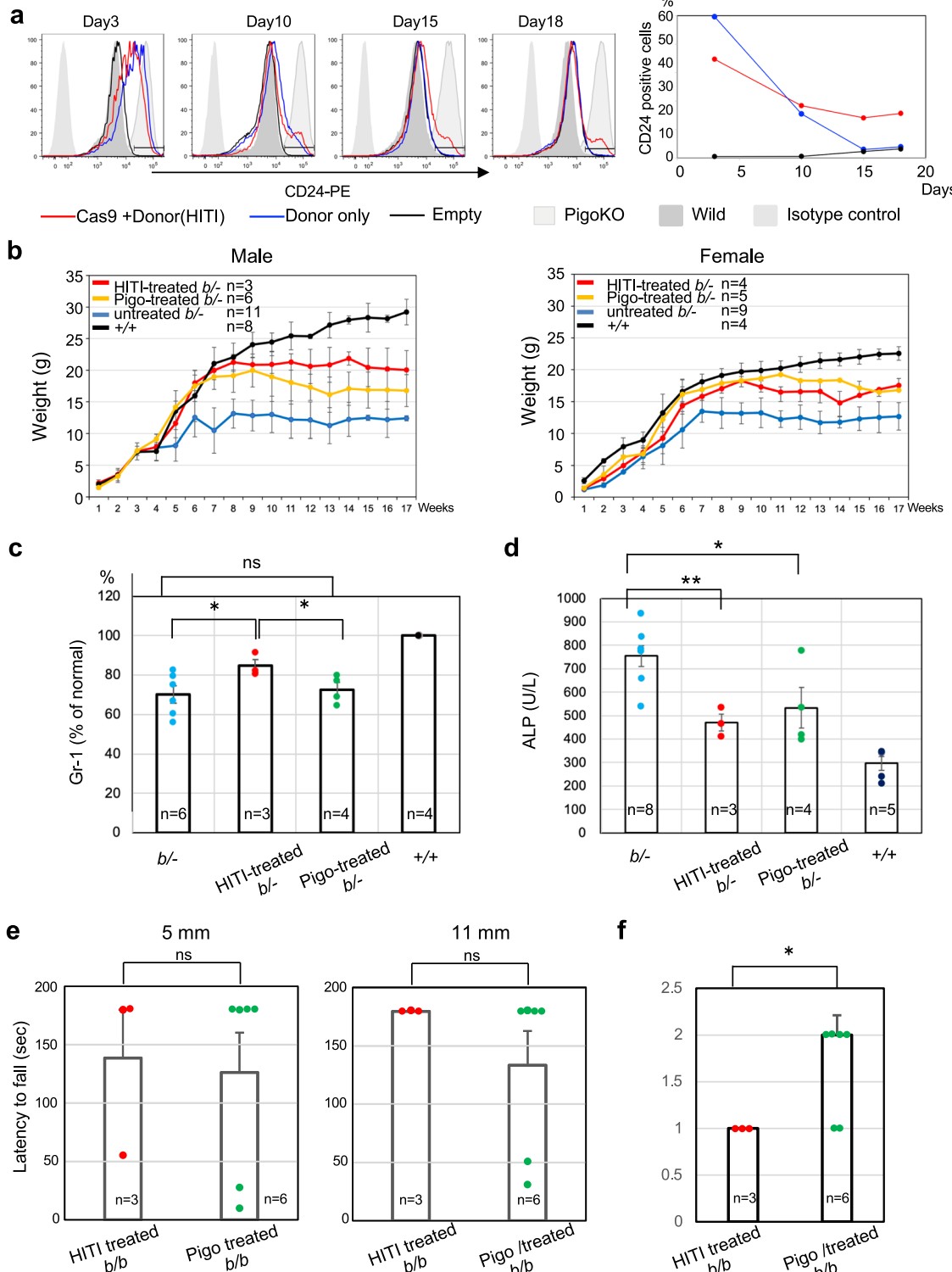

**Fig. 7 Effects of single administration of AAV donor in vitro and in vivo. a** *Pigo*-KO Neuro2a cells were transfected with pAAVCas9 and pAAV donor (pAAV-mPigoHITIgRNA) or pAAV donor only and analyzed over time by FACS for measuring the restored population. % of CD24 positive cells were shown in the right graph. **b** Comparison of growth in HITI-treated (AAVCas9 and AAV donor) and AAV donor treated *Pigo*$^{b/-}$ mice. Data are presented as mean values + SD. **c** Elevation of Gr-1, a GPI-AP, levels on blood granulocytes from *Pigo*$^{b/-}$ mice at 4 months after HITI-treated or AAV donor treated *Pigo*$^{b/-}$ mice. *n* number of animals, ns not significant. (*$p < 0.05$) (*t*-test, one-sided, *P*-value, KIKO:KIKO + HITI, 0.035; KIKO:KIKO + Pigo, 0.035) Data are presented as mean values + SEM. **d** Amelioration of hyperphosphatasia. (*$p < 0.05$; **$p < 0.01$) Data are presented as mean values + SEM. (*t*-test, two-sided, *P*-value, KIKO:KIKO + HITI, $5.5 \times 10^{-3}$; KIKO:KIKO + Pigo, 0.029). **e** Improved performance in the hanging test with 5 mm (left) and 11 mm (right) meshes of AAV donor only treated *Pigo*$^{b/b}$ mice in comparison with HITI-treated *Pigo*$^{b/b}$ mice. Data are presented as mean values + SEM. (*t*-test, two-sided). **f** Severity scores of tremor appeared in AAV donor only treated *Pigo*$^{b/b}$ mice and HITI treated *Pigo*$^{b/b}$ mice at five months (*t*-test, one-sided, *P*-value, 0.034). Data are presented as mean values + SEM. Source data are provided as a Source Data file.

cerebellum even in *Pigo*[b/−] mice (Supplementary Fig. 7a). Many kinds of GPI-AP are expressed in the central and peripheral nervous systems. For example, ephrin-A, semaphorin-7A, netrin-Gs, glypicans, and contactins play roles in axon outgrowth and navigation, whereas Nogo receptors, MAM domain-containing GPI-APs, T-cadherin, and GFRα1 regulate synaptic function and plasticity[43]. As our model mice had partial deficiency of GPI biosynthesis, these GPI-APs might be decreased to various levels. In fact, Thy1 was decreased by 40 to 50%, whereas contactin1 was not at all decreased even in the *Pigo*[b/−] mice with the severest phenotypes (Supplementary Fig. 4). It is currently difficult to speculate on which GPI-AP is responsible for the phenotype. At for the cellular level, primary neuron culture from the *Pigo*[b/b] embryo did not show any defect in neurite outgrowth or synapse numbers. In addition, the embryonic fibroblasts derived from the *Pigo*[b/−] mouse showed no decrease in GPI-AP expression (Supplementary Fig. 8).

At present, no fundamental treatment for IGD is available and the inherent monogenic nature of IGD raises the indication of gene therapy. To establish an effective treatment method for IGD, we here developed AAV-based gene therapy with genome editing by the HITI method, which allows efficient editing of the target gene even in nondividing cells like neurons[33] and lifelong effects. In case of our disease model, as the missense mutation was located in the first exon, we had to take the strategy of the target insertion of full-coding *Pigo* cDNA into the 5′UTR of the *Pigo* gene. It should be safe even if it causes indels at the target site and can achieve an endogenous expression level because the inserted *Pigo* cDNA will be regulated by the endogenous promoter and the regulatory system. Therapeutic effect was not solely by HITI-mediated integration but also by the extra-chromosomal AAV donor containing full *Pigo* cDNA because the ITR region of the AAV vector possesses weak promoter activity[35]. Indeed, the promoter activity of the ITR was comparable to the endogenous *Pigo* promoter activity in AAV donor treated mouse brain in vivo (Supplementary Fig. 11). Genomic PCR and DNA sequencing analysis of the tissues from HITI-TE-treated *Pigo*[b/b] mice revealed that the transgene was integrated in the desired genomic location, although the knock-in efficiency was ~3% at maximum in brain and ~5% in liver (Fig. 6b). Analysis of the transgene integration site by 5′ RACE also revealed that sequences of the AAV vector backbone were amplified, suggesting that *Pigo* cDNA from the extra-chromosomal AAV donor was expressed. As the *Pigo* cDNA-containing AAV donor vector alone could rescue the GPI-AP expression on *PIGO*-KO HEK293 cells when a high copy number was introduced (Fig. 3b, c), ITR-driven *Pigo* cDNA might contribute to improving the phenotypes in vivo[35]. To examine its contribution, we treated *Pigo*[b/−] mice only with AAV donor, which allows ITR-driven Pigo expression. Whereas the treatment with AAV donor only did not significantly increase Gr-1 on blood granulocytes (Fig. 7c), it reduced serum ALP levels (Fig. 7d). In the first 8–10 weeks, AAV donor only treatment had ameliorating effect on body weight similar to the HITI-TE treatment, however, the body weight of mice treated with AAV donor only tended to decline after 10 ~ 12 weeks of age (Fig. 7b). Therefore, ITR-driven Pigo expression from the AAV donor had therapeutic effects in mice. In summary, the combination of gene editing and an extra-chromosomal transgene appears to be effective to treat the IGD model mice. The same combination therapy could be designed including a promoter upstream to the sgRNA sites to sustain higher level of the transgene while enabling HITI-mediated gene repair, however we wanted to have a low *Pigo* expression in order not to affect balance between *Pigo* and *Pigg*. Pigo and Pigg are individually complexed with Pigf and mediate EthN-P transfers to Man3 and Man2, respectively. It was shown previously that PIGO overexpression affected PIGG enzyme level by interfering

generation of PIGG and PIGF complex[31]. Recently, it became clear that too much expression of transgene is harmful in gene therapy, causing liver dysfunction and thrombocythemia. Clinicians are trying to regulate the expression using drug inducible constructs. As ITR-driven expression of Pigo is at a similar level to endogenous expression, it would be a reasonable choice in combination with gene editing therapy. Further analysis is required to investigate whether our combined strategy has an advantage over conventional gene transfer, even if it has a risk of off-target effects. In fact, off-target effects could be detected in 9% of liver transcripts analyzed by 5′ RACE.

It is interesting that the neurological phenotype of HITI-TE-treated *Pigo*[b/b] mice was clearly improved despite the low efficiency of precise genome editing, which was lower than 10%. Similar findings were reported in previous papers, showing that low gene correction efficiency was sufficient to improve the phenotypic outcomes[33]. The authors explained that an indirect non-cell autonomous effect from the small number of gene-corrected cells in the tissue improved the phenotype. There are several lines of evidence published showing that GPI-anchored proteins (GPI-APs) are transferred to the other cells. GPI-APs expressed on the surface of transgenic mouse red blood cells were transferred in a functional form to endothelial cells in vivo[44]. Piga[−] embryoid bodies (EB) produced no secondary hematopoietic colonies, however, in the chimeric EB composed with normal cells and Piga[-] cells, hematopoiesis from knock-out cells was reconstituted due to transfer of GPI-APs from normal to knock-out cells[45]. GPI-APs are efficiently sorted in exosomes and are transferred to the other cells[46,47]. According to these lines of evidence, GPI-APs transfer from the gene-corrected cells might contribute to the phenotype improvement. Restoration of Gr-1 on granulocytes was seen at 4 months after the HITI-TE-treatment, suggesting gene-editing event in hematopoietic progenitor cells.

Our results proved that gene therapy for IGD is a potential therapeutic strategy. They also suggested that the symptoms of IGD are reversible if treated at a young age. To further promote the practical use of this approach, there is a need to develop optimized gene therapy for humans. AAV-PHP.eB can penetrate diffusely into the mouse brain even in adults, but only for limited strains[48,49], while it does not work in humans. Recently, Ly6A has been reported as an endothelial receptor that allows AAV-PHP.eB to cross the blood–brain barrier (BBB)[49,50]. The lymphocyte antigen-6 (Ly6)/urokinase-type plasminogen activator receptor (uPAR) proteins form a large superfamily of structurally related proteins, which are all GPI-APs. Therefore, *Pigo* KI mice are expected to express less Ly6A than wild-type mice and are at a disadvantage for AAV-PHP.eB treatment. The expression levels of Ly6A were also found to be lower in newborn mice than in adult mice[51]. Nevertheless, there was a clear phenotypic improvement, suggesting that the expression level of Ly6A in the affected newborn mice was sufficient for the virus to cross the BBB. The human and mouse genomes contain 35 and 61 Ly6/uPAR family members, respectively, but there is no ortholog of Ly6A in primates[52]. Other Ly6 proteins expressed on the endothelium of the primate central nervous system can be candidate receptors. Developing capsids that bind to these receptors should open the way for efficient gene therapy of various inherited neurological diseases.

## Methods

**Functional analysis of the *Pigo* mutant cDNAs in the mouse cells.** Clones of *Pigo*-knockout (KO) Neuro2a, a mouse neuroblastoma cell line, were generated by CRISPR–Cas9 system. Each *Pigo* mutant cDNA (R119W, T130N, K1051E) driven by a strong promoter, SRα, was transfected into *Pigo*-KO Neuro2a cells by Lipofectamin 2000 (Invitrogen) together with luciferase expressing plasmid for the transfection control. Two days later, cells were stained with FLAER (fluorescent-labeled inactive toxin aerolysin, CEDARLANE), which reacts with GPI-APs, or PE

labeled anti-CD24 antibody (Biolegend) and were analyzed by flow cytometry (MACSQuant Analyzer; Miltenyi Biotec) with Flowjo software (Tommy Digital). Lysate of these transfectants were subjected to SDS PAGE and immunoblotting for HA tag. For quantification, relative protein expression was determined by dividing HA-Pigo band intensities (detected by rabbit monoclonal anti-HA, 1:2000 dilution, C29F4, 3724 S, Cell Signaling) by those of GAPDH (dected by mouse monoclonal anti-GAPDH, 1:2000 dilution, AM4300, Invitrogen) and by Luciferase activity to normalize for both loading and transfection efficiencies and resulting values converted to make the wild-type value as 1.

**Animal generation.** Three lines of KI ES cells bearing different *Pigo* mutations (A-line, R119W; B-line, T130N; C-line, K1051E) were established using CRISPR–Cas9 system (Supplementary Fig. 2). In brief, C57BL/6 derived ES cells were co-transfected with px459 vector (Addgene) containing a target gRNA sequence (ACGACCACCATGCAGCGTCT for A-line, GCTGTACCGATCT-CAGGTGG for B-line, AAGTAAGTGGTTTCCCGTGA for C-line) and a donor plasmid containing a 1.5 kb homologous region for efficient target recombination, which includes a pathogenic mutation (c.355 C > T in nucleotide sequence for A-line, c.389 C > A for B-line, and c.3151 A > G for C-line), with silent one-base substitution mutations to confer resistance against Cas9 cleavage, and a restriction enzyme site for easy genotyping, (SalI site for A- and B-line for, FspI site for C-line). Successful KI ES cells were picked. Three lines of mice were generated according to the usual method using these KI ES cells bearing A, B, or C-line mutation. An ES clone bearing a *Pigo*-KO allele was accidentally generated during A-line KI process due to a homologous recombination failure. The KO allele contains two repeated exon 1 with intervening sequences, which is transcribed to tandemly connected exon 1, generating a premature termination codon, leading to nonsense mediated decay.

**Animals.** All mice were C57BL/6 background and maintained in SPF under a 12 h light /12 h dark cycle with food and water provided ad libitum. Temperature and humidity were within the recommended range (20 ~ 24° and 40 ~ 60%, respectively). All animal procedures were approved by the Animal Care and Use Committee of the Research Institute for Microbial Diseases, Osaka University, Japan. The methods were carried out in accordance with the approved guidelines. *Pigo* heterozygous KI mice and KO mice were maintained by mating with C57BL/6 mice.

**Genotyping.** The *Pigo* A- and B-line KI mice were genotyped by polymerase chain reaction (PCR) using PrimeSTAR GXL DNA polymerase (Takara bio) with the primers 5′ TTGCCACCCTGGAAATGTTG3′ (primer 1) and 5′ TAGAGGTGTTCCAAGATGCCG3′ (primer 2). Then, PCR products were digested with SalI-HF (New England Biolabs) and separated on a 1% agarose gel. KI alleles were detected by the digested PCR product bands of 1021 bp and 761 bp. The *Pigo* C-line KI mice were also genotyped by PCR with the primers 5′ CCGGCTCAGAGTTTTCTCATT3′ (primer 3) and 5′GAAACA-TAGTGCTTCAAACTGTG3′ (primer 4), followed by digestion with FspI (New England Biolabs). KI alleles were detected by the digested PCR product bands of 831 bp and 665 bp. The KO allele was detected by PCR with the primer 1 and the primer 5′ GAAAGGAGCGGGCGCTAGGG3′ (primer 5), followed by digestion with SalI-HF, generating the bands of 878 bp and 761 bp.

**Measuring alkaline phosphatase (ALP) activity.** Whole blood was collected from the mouse tail artery, and plasma was separated by centrifugation. ALP activity in plasma was measured in the clinical laboratory by the JSCC recommended method using p-Nitrophenylphosphate (pNPP) as a substrate. As ALP activity is high at an early age, mice older than 4 months were investigated.

**Flow cytometry of mouse blood cells.** After separating plasma from whole blood by centrifugation, the red blood cells were lysed to assess the surface expression of GPI-AP on granulocytes. Blood cells were stained with Pacific Blue$^{TM}$ conjugated anti-mouse Ly-6G/Ly-6C (Gr-1) antibody (1:100 dilution, RB6-8C5 #108430 Biolegend) and Alexa488 conjugated anti-CD48 antibody (1:100 dilution, HM48-1, #103414 Biolegend) and the gated granulocytes were analyzed by flow cytometry (MACSQuant Analyzer; Miltenyi Biotec, Bergisch Gladbach, Germany) with Flowjo software (Tommy Digital, Tokyo, Japan).

**Video EEG recordings and analysis.** Adult mice at 4 months of age or older were used for all EEG recordings. The mice were anaesthetized with isoflurane and implanted with EEG electrodes (Pinnacle 8201: 2 EEG/1 EMG Mouse Headmount) according to the manufacturer's instructions. The electrodes for the EEG recordings were positioned over the frontal and occipital cortex (frontal: 2 mm anterior to bregma, 2 mm lateral to midline; occipital: 2 mm anterior to lambda, 2 mm lateral to midline). The headmounts were held in place with dental cement. After the cement set, the electrodes were attached to thin cables linked to a computer running software, allowing visualization of the EEG activity with simultaneous video recording (Vital Recorder, Kissei Comtec). Video EEGs were recorded overnight, and the files were reviewed for background activity, epileptic discharge,

and seizure activity (SleepSign, Kissei Comtec). Additionally, low dose of pentylenetetrazole (PTZ) (20 mg/kg) was administered intraperitoneally, and video EEGs were recorded for 2 h to evaluate PTZ-induced seizure susceptibility using Modified Racine scale[37]. Frequency and amplitude of 8 h recording EEG data were calculated by Fast Fourier transform (FFT) power spectral analysis[53]. Data were analyzed in 10-s epochs by the SleepSign software. The EEG signal was separated into five regions per epoch. Each region was FFT calculated by using 256 datum points (2 s) before the five spectra were averaged. The spectrum has the resolution of 0.5 Hz.

**Magnetic resonance imaging.** In vivo magnetic resonance imaging (MRI) of mice were conducted using an 11.7 T vertical bore scanner (AVANCE II 500WB; Bruker BioSpin, Ettlingen, Germany). Two males and one female of *Pigo*$^{b/-}$ mice (15 weeks) were used. *Pigo* heterozygous KO littermates of each male and female mice were also used for comparison (two males and one female). Anesthesia of mice was initially induced with 2% isoflurane and maintained with 1.6% isoflurane during MRI. Body temperatures of mice were maintained at 37 °C with circulating warm water. $T_2$ weighted images were obtained by the Rapid Acquisition with Relaxation Enhancement (RARE) technique[54]. Acquisition parameters were as follows: field of view [FOV] = 20 mm × 20 mm, matrix size = 256 × 256, in plane resolution = 78 μm, slice thickness = 300 μm, repetition time [TR] = 6000 ms, echo time [TE] = 37.6 ms, number of averages [NA] = 12, acquisition time = 19 min.

**Western blotting.** The brains of adult mice at 4 months of age were taken after perfusion with saline, homogenized and solubilized with 60 mM of n-octyl-β-D-glucoside containing lysis buffer, followed by centrifugation to remove the debris. After measuring protein amount by BCA assay, lysates were applied to SDS-PAGE followed by western blotting. Primary antibodies used for the assay were goat anti-human contactin1 (1:1000 dilution, AF904, R&D systems), rat monoclonal anti-mouse Thy1 (1:500 dilution, 105202, G7, BioLegend), rat monoclonal anti-MBP (1:3000 dilution, clone12, MCA409S, Bio-Rad), and mouse monoclonal anti-GAPDH (1:2000 dilution, AM4300, Invitrogen). Secondary antibodies used were HRP-conjugated anti-goat or rat or mouse IgG-HRP.

**Histological analysis of the mouse brain.** The mice at 4 months of age were deeply anesthetized with Isoflurane and transcardially perfused with 20 ml of PBS/1% heparin followed by 10 ml of 4% paraformaldehyde (PFA)/PBS. The brains were taken and fixed in 4% PFA/PBS in 4 °C for an overnight and brains were washed in PBS. For paraffin-embedded sections, brains were cut and soaked in 50 and 70% ethanol and xylene sequentially by Thermo Scientific STP 120. Paraffin infiltrated brains were embedded in paraffin and were sectioned using paraffin microtome. After deparaffinized in xylene, sections were soaked in 100%, 95 and 70% ethanol sequentially. These brain sections were washed in H$_2$O and soaked in Cresyl Violet acetate solution for 10 min and washed in 100% ethanol and embedded in the mounting solution (Nissle staining). For making the frozen section, brains were dipped into 10% sucrose/PBS for an overnight followed by 20% sucrose/PBS for 2 days. After cut into blocks, brains were equilibrated with PBS, embedded into compound (FSC22, Leica) and frozen in the liquid N$_2$. Frozen blocks were sectioned using CryoStar NX50H (ThermoFisher). For immunohistochemical staining, sections on the slide glasses were dried up for 30 min and treated with 0.1% TritonX-100 in PBS for 5 min and washed with PBS. They were treated with blocking solution (0.1% TritonX-100 PBS/1% donkey serum/1xBlockAce (DS Pharma) for 1 h), followed by staining with the first antibody in the blocking solution for 2 h at room temperature. After washing with 0.1% TritonX-100 in PBS, sections were stained with the second antibody in the blocking solution for 1 h at room temperature. After washing with 0.1% TritonX-100 in PBS and PBS sequentially, DAPI staining was performed for 5 min, washed and embedded in the mounting solution (Prolong Diamond, ThermoFisher).

Sections were analyzed by the microscope (KEYENCE BZ-X800). The first antibodies used were mouse monoclonal IgM O4 (1:250 dilution MAB1326 R&D), rat monoclonal anti-MBP (1:250 dilution, clone12 MCA409S BIO-RAD), rabbit monoclonal anti-Contactin2 (1:100 dilution, EPR5106, ab133498 Abcam), rabbit polyclonal anti-Synapsin (1:250 dilution, 106103, Synaptic System); the second antibodies were goat anti-mouse IgM Alexa488 (1:500 dilution, A21042, Invitrogen), goat anti-rat IgG Rhodamin (1:500 dilution, AP183R, Chemicon), donkey anti-rabbit IgG FITC (1:200 dilution, 711-096-152, Jackson). For quantitation, area (for O4), cell numbers (for GFAP and Contactin2), dot numbers (for Synapsin) within a certain area were determined using Hybrid Cell Count of KEYENCE BZ-X800 analyzer.

**Culturing primary neurons from mouse hippocampus.** E17 fetuses were taken, and the heads were decapitated by scissors and immediately transferred to ice-cold PBS stored on ice. Brain dissection was performed in 10-ml Hanks's solution on a 60-mm Petri dish on ice.

Hippocampi were incubated in 300 μl of Hanks's solution containing papain (PDS2, Worthington) for 30 min at 37 °C. 600 μl of Neurobasal medium (21103-049, Life technologies) with 10% FCS was added to stop the enzymatic dispersion reaction and dissociated by gently pipetting. After filtrated with cell strainer 70 μm (BD), centrifuged and suspended with Neurobasal medium with B27 supplement

(17504-044, Life technologies) and Glutamax (35050-061, Invitrogen). Cells were seeded in the poly-L-Lysine coated 96 well plates ($1.0 \times 10^4$ cells/100ul/well MS-0096L SUMITOMO). On day2 and day7, cells were fixed with 4% PFA for 15 min at room temperature, washed with PBS, permeabilized with 0.25% Triton-X100 for 5 min at room temperature, washed with PBS three times and then, incubated with blocking buffer (Nakarai) for 30 min at room temperature. Cells were incubated with the first antibody for overnight at 4 °C, washed with PBS and incubated with the second antibody for 2 h at room temperature. After washed with PBS, cells were analyzed with confocal microscope, Opera LX (PerkinElmer). The first antibodies, mouse monoclonal anti-Nestin (1:300 dilution, Rat401, ab6142, Abcam), rabbit polyclonal anti-β-tubulin (Tuj1) (1:2000 dilution, ab18207, Abcam), chicken polyclonal anti-MAP2 (1:5000 dilution, ab5392, Abcam), rabbit monoclonal anti-NeuN (13E6, ab177487, Abcam), guinea pig polyclonal anti-Synapsin 1 (1:1000 dilution, 106 004, Synaptic systems), mouse monoclonal anti-Tau-1 (1:500 dilution, PC1C6, MAB3420, Chemicon); the second antibodies, goat anti-mouse IgG-Alexa488 (1:500 dilution, A11029, Life Technologies), goat anti-chickenIgY-Alexa555 (1:500 dilution, A32932, Life Technologies), goat anti-rabbit-Alexa647 (1:500 dilution, A32733, Life Technologies), goat anti-guinea pig IgG Alexa488 (1:500 dilution, A-11073, Life Technologies).

**Animal behavioral analysis.** Total of 17 male and 19 female of *Pigo* A-line mice (5–9 months of age) were sent to the animal facility of Asubio Pharma Co., Ltd. and behavioral tests were performed.

Rotarod tests were performed to assess motor coordination and balance. The subject mice were trained for 2 min at 4 rpm and then tested on the accelerated setting (4–40 rpm, 5 min) two times, and the average of the latency to fall was recorded.

Novel object recognition test was performed in the square chamber to test learning and memory in mice. It consists of three sessions, one habituation session, one training session, which involves visual exploration of two identical objects and one test session, which involves replacing one of the previously explored objects with a novel object. The use of each set of object is counterbalanced so that each object is used equally as a familiar object and as a novel object. Because mice have an innate preference for novelty, a mouse that remembers the familiar object will spend more time exploring the novel object. Using Video tracking system (ANY-maze version 4.72 software; Stoelting Co.), exploring time was measured.

Muscle weakness and coordination deficit were measured by four-limb hanging test. The latency to fall was recorded with a 300 s cut-off time. The distance traveled by the mouse before falling off the grids was also recorded using video tracking system. The performance of the mutants was compared with that of wild-type controls.

As for *Pigo* B-line mice treated with AAV, we assessed the severity of tremor using a five-score scale (Supplementary Movie 2), and four-limb hanging test (5 mm and 11 mm mesh) was performed at 5 months of age to demonstrate neuromuscular impairment and motor coordination. Tremor was analyzed weekly, and the severity of tremor was evaluated using a five-score scale by one observer, and the score was checked by another observer by watching video.

**Generation of genome editing AAV plasmids.** AAV-nEFCas9 plasmid was purchased from Addgene (#87115). To construct a donor/gRNA AAV for HITI, donor DNA sandwiched by Cas9/gRNA target sequences and a gRNA expression cassette were subcloned between ITRs of PX552 (Addgene #60958), and generated pAAV-mPigoHITIgRNA. *Pigo* starts from exon 1 and the mutations are located on this exon, thus, insertion of *Pigo* full-coding cDNA in 5′UTR can produce a normal transcript rather than local transcript including the target mutations in *Pigo* gene. Whole *mPigo* cDNA, including 5′UTR, coding region, and 3′UTR, and downstream genomic sequence were flanked by Cas9/gRNA target sequences, which is expected to integrate within 5′UTR of exon 1 of *mPigo* by HITI. AAVs were packaged with PHP.eB capsid[38] by helper-free triple transfection procedure, which were purified by cesium chloride density gradient ultracentrifugation. Viral titer was determined by quantitative PCR using Taq-Man technology (Life Technologies).

**Verification of the effect of HITI method in vitro.** pAAV-nEFCas9 and pAAV-mPigoHITIgRNA were co-transfected to *Pigo*-KO Neuro2a cells and two days later, cells were analyzed for the surface expression of GPI-AP, CD24 by FACS. Their genomic DNA was extracted using Wizard™ Genomic DNA Purification Kits (Promega) according to manufacturer's instructions. The HITI-mediated gene knock-in locus was amplified with PrimeSTAR GXL DNA polymerase (#R050A, Takara Bio USA, Inc.). The 5′ junction site was amplified by PCR with following HITI-specific primer: mPigoHITI5′-F1 (5′-TGGTCCCTGGGCTTTTCTTCT TTCATGCTT-3′) and mPigoHITI5′-R1 (5′-AGGTATGAAGAAACCGGGACA CCTGCT-3′) were used. The 3′ junction site was amplified by PCR with following HITI-specific primers: mPigoHITI3′-F1 (5′-ATGGTCTCGGAAGGTGTTTGC TCCCAAGTTCA-3′) and mPigoHITI3′-R1 (5′-GAACCCGGCCAACACCT GGACATGC-3′) (Supplementary Table 2). PCR products were cloned into the pCR-Blunt II-TOPO vector with Zero Blunt TOPO cloning kit (#450245, Invitrogen). Amplicons were sequenced using an ABI 3730xl sequencer (Applied Biosystems). As for the FACS analysis over time after transfection in Fig. 7, *Pigo*-

KO Neuro2a cells were co-transfected with pAAV-nEFCas9, pAAV-mPigoHITIgRNA (HITI-TE) and pME-BFP or pAAV-mPigoHITIgRNA (pAAV donor only) and pME-BFP. 3days later, BFP positive cells were sorted and analyzed by FACS.

**Intravenous AAV injection to newborn *Pigo* deficiency mouse model.** The newborn (P1–P3) B-line homozygous KI or KIKO mice were subjected to intravenous AAV injection as in a previous report[55]. Prior to procedure pups were incubated on ice for 1 min to anesthetize and subsequently injected via the temporal vein using a 30 G insulin syringe with needle (BD lo-dose Insulin syringe with needle, 30 G, 1/2 ml). Then, the pups were given 2–3 min to rewarm and recover and returned to the cage.

**Evaluation of in vivo HITI events in HITI-TE-treated mice.** To examine HITI-mediated knock-in event by Sanger sequence, genomic DNA from brain and liver was extracted and the HITI-mediated gene knock-in locus was amplified in the same way as mentioned above. The 5′ junction site was amplified by PCR with following HITI-specific primers in liver: mPigoHITI5′-F1 and mPigoHITI5′-R1 were used. The 5′ junction site was amplified by nested PCR with following HITI-specific primers in brain. For 1st PCR, mPigoHITI5′-F1 and mPigoHITI5′-R1 were used. For 2nd PCR, mPigoHITI5′-F2 (5′-CGAGCCCTGCCGCTGCACTTCCG-3′) and mPigoHITI5′-R2 (5′-ACATCCCACGAACCGCCGTCCAGGGTAG-3′) were used (Supplementary Table 2). The 3′ junction site was amplified by PCR with following HITI-specific primers: mPigoHITI3′-F1 and mPigoHITI3′-R1. PCR products were cloned into the pCR-Blunt II-TOPO vector with Zero Blunt TOPO cloning kit (#450245, Invitrogen). Amplicons were sequenced using an ABI 3730xl sequencer (Applied Biosystems).

**5′RACE-based off-target and transcript analyses.** SMARTer RACE 5′/3′ Kit (#634858, Takara Bio USA, Inc.) was used for performing the 5′- rapid amplification of cDNA ends (5′RACE) according to manufacturer's instructions. 1 μg total RNA was used for this reaction. *Pigo* exon 2/3 junction-specific primer used in this experiment was 5′-GATTACGCCAAGCTTA-CATCCCACGAACCGCCGTCCAGGGTAG-3′ (primer 10) for 1st PCR. PCR products were cloned into the In-Fusion HD Cloning Kit. RACE fragments were sequenced using an ABI 3730xl sequencer (Applied Biosystems). The captured exons which are located to upstream of Pigo exon 1 were mapped on UCSC mouse genome browser (NCBI37/mm9) (https://genome.ucsc.edu/cgi-bin/hgGateway?db=mm9).

**Analysis of promoter activity of ITR which drives *Pigo* in AAV donor.** PIGO-KO HEK293 cells[22] were transfected with an AAV donor plasmid (pAAV-PHP.eB-mPigoHITIgRNA) or a strong promotor (SRα) driven *mPigo* (pME mPigo) by Lipofectamin 2000 (Invitrogen). They were also infected with AAV donor or AAV nEF-GFP at 100 K MOI in the serum free medium, pretreated with 100 mU/ml of neuraminidase. After 24 h, the serum free medium was changed to medium containing 10% FCS. Two days later, cells were stained with anti-CD59 (5H8), followed by PE labeled anti-mouse IgG and were analyzed by flow cytometry (MACSQuant Analyzer; Miltenyi Biotec) with Flowjo software (Tommy Digital).

$2 \times 10^5$ /well of NIH3T3 cells, derived from a murine embryo fibroblast cell line, were plated in the 12 well plate. On the next day, they were pretreated with 100 mU/ml of neuraminidase in the serum free medium for 2 h at 37°C, after washing with serum free medium, cells were infected at various MOI with AAV-PHP.eB-nEF-EGFP or AAV donor (AAV-PHP.eB-mPigoHITIgRNA). After 48 h from infection, cells were harvested, FACS analysis was performed for AAV-PHP.eB-nEF-EGFP infected cells, genomic DNA and RNA were isolated from AAV donor infected cells. Total RNA was reverse transcribed using Superscript VILO kit (Invitrogen) and qPCR were performed with the cDNA and genomic DNA by SYBR Green PCR mater mix (Applied Biosystems) and primers using StepOnePlus Real-Time PCR Systems (Applied Biosystems). *Pigo* expression was determined by qPCR with the comparative Ct method using the primer GCAGTAACTTTGCCAGCCATGC (qPCRmPigo-F1) and CACTGTGTGCAGGTCTCTGAC (qPCRmPigo-R1). For endogenous control, primer TATGACCCCTATCACTTCTG (qPCRmTbp-F1) and TTCTTCACTCTTGGCTCCTGT (qPCRmTbp-R1) (for expression of *Tbp*, TATA-box binding protein) were used (Supplementary Table 2). Viral gene copy number was determined by qPCR using the same primer qPCRmPigo-F1 and qPCRmPigo-R1. Standard curve was prepared as a function of dilution of AAV donor plasmids. Increase of *Pigo* expression and *Pigo* gene copy number from non-infected cells corresponds the viral derived expression and gene copy number, respectively. At MOI $10^3$, ITR-driven expression and viral gene copy number was 0.82 and $0.24 \times 10^6$, respectively, while in non-infected cells, *Pigo* expression and gene copy number was 1 and $0.003 \times 10^6$, respectively. Relative ITR vs endogenous promoter activities were 0.82/0.24 vs 1/0.003, ie 3.4 vs 333.

For in vivo analysis, wild-type mice ($n = 3$) were treated with AAV donor on 2 days after birth. Two weeks later, AAV treated mice and non-treated littermates ($n = 3$) were sacrificed and brains were taken out. Total RNA and genomic DNA were isolated simultaneously with Allprep DNA/RNA mini kit (Qiagen) from the whole cerebrum. Total RNA was reverse transcribed using Superscript VILO kit (Invitrogen) and qPCR were performed with the cDNA and genomic DNA by

SYBR Green PCR mater mix (Applied Biosystems) and primers using StepOnePlus Real-Time PCR Systems (Applied Biosystems). Standard curve was made by serial dilutions of AAV donor plasmids in the solution containing Herring sperm DNA (Promega). For ITR-driven expression, the qPCR probe was set in U6 promoter. For endogenous *Pigo* expression, the probe was set in the *Pigo* exon boundary (Supplementary Table 2). cDNA copy numbers were normalized with relative *Tbp* expression as an endogenous control. ITR promoter activity is calculated by *U6* cDNA/*U6* genome copy numbers, whereas endogenous *Pigo* promoter activity is calculated by *Pigo* cDNA copy/*Pigo* genome copy numbers.

**Reporting summary**. Further information on research design is available in the Nature Research Reporting Summary linked to this article.

## Data availability

Full scans for all western blots and raw data are provided in the Supplementary Information. All other data are available from the corresponding author on reasonable request. Source data are provided with this paper.

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

## Acknowledgements

We thank Junji Takeda (Osaka University) for discussion, Yoshihiro Tani (Asubio Pharma Co.) for performing mice behavioral analysis, Keiko Kinoshita, Saori Umeshita, Kae Imanishi, Kana Miyanagi (Osaka University), and Maki Fujiwara (Kyoto University) for technical help. We thank Edanz Group (https://en-author-services.edanzgroup.com/ac) for editing the English text of a draft of this manuscript. This work was supported by JSPS and MEXT KAKENHI grants (JP16H04753 and JP17H06422 for T. Kinoshita), a grant from Ministry of Health, Labor and Welfare (20FC1025) and Mizutani Foundation for Glycoscience, KOSE Cosmetology Research Foundation, the Osaka Medical Research Foundation for Intractable Diseases, a grant from Practical Research Project for Rare/Intractable Diseases from the Japan Agency for Medical Research and Development (AMED) (21ek0109418h0003 for Y. Murakami), grants form JSPS KAKENHI (18H04036) and Takeda Science Foundation for K. Suzuki and a grant from AMED(JP20dm0307021) for K. Inoue.

## Author contributions

R.K., K.S., T.K., K.O. and Y.M. designed the study, R.K., K.S., J.N., E.A., Y.Y., Y.S., S.N. and Y.M. acquired the data and conducted experiments. MI made the knock-in mice. K.I. prepared AAV. R.K., K.O., S.N., K.S., T.K. and Y.M. wrote and edited the paper.

## Competing interests

The authors declare no competing interests.
