## [Peer Review File · Nature Communications]

Reviewers' Comments:

Reviewer #1:

Remarks to the Author:

Kuwayama and colleagues studied inherited glycosylphosphatidylinositol (GPI) deficiency (IGD) through the generation and the characterization of KI and KO mutant mouse models for the *Pigo* gene, a key factor for GPI-anchor-synthesis pathway. In the second part of the manuscript, authors exploited these models to set a novel gene therapy approach aimed to restore wild-type *Pigo* expression through a combinational strategy. This method is based on *Pigo* transgene expression triggered by the ITR of an AAV vector that works at the same time as a donor DNA to repair the endogenous gene through the HITI strategy.

The first part of the manuscript presents the highest novelty, nevertheless authors missed to investigate some important aspects regarding the characterization of *Pigo* mutant KI mouse lines. In fact, their analysis is mainly macroscopic focusing on overall brain morphometry, epilepsy and few behavioral tasks. It would be very informative to perform histological and cellular analyses to determine levels of cell-death, cellular alterations and functional impairments in *PIGO* KI mice. Moreover, it is not shown which is the effect of these mutations on *Pigo* expression at mRNA and protein level. This is a crucial analysis to understand, for instance, why the T130N mutation (*Pigob/b*) is associated with a more severe pathological phenotype in comparison with the other two mutations. Moreover, as stated by the authors in the discussion [In human cells, Arg119Trp mutant showed the most severe decrease in activity leading to the severest clinical symptoms, such as intractable seizures, severe intellectual disability, and multiple organ anomalies, whereas, in the murine cells, the severest case was the Thr130Asn mutant], there is a difference between mouse and human cells regarding the effect of these mutations. Thus, for example, a more detailed characterization through RT-PCRs and Western blot analyses is needed to solve this discrepancy.

The second part remains more problematic and less than conclusive. Based on the presented data, it appears that the HITI-mediated integration of WT *Pigo* cDNA upstream the endogenous mutant gene poorly contributes in ameliorating the pathological phenotype and probably is responsible for the aberrant mRNA products observed in the liver. Indeed, as shown in Fig. 6b only a minimal proportion of the corrected *Pigo* transcripts results from the donor integration (< 3%), with a more preponderant contribution from the ITR-mediated AAV transcription. At this point, the exact contribution of the donor integration in the phenotypic rescue needs to be stringently ascertained. For this, authors need to test both in *Pigo*-KO Neuro2a cells and in *Pigob/b* mice the overall effects (gene expression and mouse behavior) of the donor DNA (pAAV m*Pigo*HITIGRNA) without the Cas9. This is the most rigorous way to ascertain whether the Cas9-dependent HITI-mediated integration significantly contributes or not to the pathological phenotypic rescue.

Another troublesome aspect of this study is related to the only use of ITRs to express the therapeutic gene. The promoter activity of the ITR is poorly studied and its cell-type specificity and temporal dynamics is unknown. Moreover, authors did not compare *Pigo* transgenic expression triggered by the ITRs with a wild-type control. Thus, it has not been assessed if AAV-mediated *Pigo* expression resembles its physiological expression in time and space. Detailed analyses of ITR-dependent activity in different cell types and tissues and its temporal evolution need to be described in detail to support its potential therapeutic validity.

Minor revision:

- 1) In Figure 1 a Sanger sequence interferogram is required to show the presence of the mutations
- 2) In Figure 3 the efficiency (on-target analysis) and the selectivity (off-target analysis) of the selected sgRNA should be included
- 3) In Figure 5 the 5' junction PCR in brain should be included
- 4) In Extended Figure 3 a graph representing the Western blot quantification should be included
- 5) Behavioral tests are presented only for the A-line KI mice while it should be presented also for the other 2 lines.

Reviewer #2:

Remarks to the Author:

This is an impressive study aimed at developing and treating a mouse model of the human genetic disorder, *PIGO* deficiency. The authors accomplish this and provide important insights into 1.)

identifying the best mutation to analyze and 2.) using cleverly designed integrating and extra chromosomal AAV-based vectors of the normal allele. This well written presentation offers an approach which could be generalized to study and treat other rare diseases where multiple mutations in a single gene present challenges of both phenotype and rescue strategy.

Previously described knockout PIGO animals are nonviable and any useful model would require knockin the corresponding mutations. The authors create and evaluate the impact of 3 knockin mutations (a, b, and c) using cell based assays and then create 3 mouse lines carrying each homozygous mutation or as a heterozygote with the null allele, e.g., b/- (T130N/-). Each of these lines was extensively evaluated for growth, neurological, structural, behavioral, and biochemical phenotypes. There was a graded response depending on the mutations and whether they were homozygous b/b or b/-. The extensive phenotyping provides a solid genotype-phenotype correlation. Only the severest mutation (b/- and b/b) recapitulated patient PIGO phenotype as well demonstrated in the videos of b/b mice.

Treatment by gene replacement therapy using rAAV is possible, but the need to treat very young patients means that the extra chromosomal normal allele would be diluted in time. Furthermore, PIGO forms a complex with another protein (PIGF) which also interacts with PIGG. Unregulated over expression of PIGO could upset the appropriate distribution of PIGF. Ideally, integrating the rescuing gene under the control of its endogenous promoter would be preferred. To this end, the authors applied their previously published non-homologous end joining (NHEJ)-based homology-independent strategy for targeted transgene integration (HITI) giving higher integration into dividing and non-dividing cells. They combined this with an extra chromosomal weak promoter of the AAV vector to keep expression to a modest level.

The single intravenous dose 100 billion genome copies of the therapeutic vectors into newborn of b/- and b/b mice had significant beneficial effects in both strains in terms of increased appearance of GPI-anchored protein (Gr-1) on granulocytes, decrease of plasma alkaline phosphatase indicating return of functional GPI synthesis. Low body weight and slow growth also improved. Tremors, grip strength and hindlimb clasping, and seizures also improved with the therapy. The extrachromosomal AAV appeared to account for most (60-80%) of the transcripts in the brain and only 3-5% from the integrated version. At this point the off-target effects appeared modest

Overall this is a novel and compelling paper about the therapeutic potential of a gene therapy for PIGO deficient patients. While more work is needed before this is ready for human trials, it provides a solid foundation and an exciting step forward.

My only suggestion is to complement the extensive genotype phenotype studies with a biochemical assay of PIGO activity. While this is not essential, it would provide a perspective of phenotype and residual activity. As more PIGO mutations are identified, functional assay data might provide a database of likely disease course. Perhaps a simple TLC plate analysis could do this.

REVIEWER COMMENTS

Reviewer #1 (Remarks to the Author):

Kuwayama and colleagues studied inherited glycosylphosphatidylinositol (GPI) deficiency (IGD) through the generation and the characterization of KI and KO mutant mouse models for the *Pigo* gene, a key factor for GPI-anchor-synthesis pathway. In the second part of the manuscript, authors exploited these models to set a novel gene therapy approach aimed to restore wild-type *Pigo* expression through a combinational strategy. This method is based on *Pigo* transgene expression triggered by the ITR of an AAV vector that works at the same time as a donor DNA to repair the endogenous gene through the HITI strategy.

The first part of the manuscript presents the highest novelty, nevertheless authors missed to investigate some important aspects regarding the characterization of *Pigo* mutant KI mouse lines. In fact, their analysis is mainly macroscopic focusing on overall brain morphometry, epilepsy and few behavioral tasks.

- *It would be very informative to perform histological and cellular analyses to determine levels of cell-death, cellular alterations and functional impairments in PIGO KI mice.*

Thank you for pointing out. In this revised version, we added Supplementary Fig. 7 and Fig. 8. The supplementary Fig. 7 shows the Nissle staining as well as immunofluorescent staining of brain sections from *Pigo*^{b/-}(KI/KO) mice compared to the *Pigo*^{+/-}(KO hetero) mice. As we mentioned in the discussion, we didn't find any abnormalities in them. We added the sentences explaining these data in the Results. We also cultured the primary neurons from the fetus. As can be seen in the Supplementary Fig. 8a and 8b, the cells from the *Pigo*^{b/b}(KI homo) fetus had defect in attachment to the dish, however, after attachment, they developed normally. We established the mouse embryonic fibroblast (MEF) from the *Pigo*^{b/-}(KI/KO) and control mice. GPI-AP stained by FLAER was not decreased compared to the controls (Supplementary Fig. 8c). In contrast, we could see the significant decrease in Gr-1, a GPI-AP on the granulocytes from KI homo and *Pigo*^{b/-}(KI/KO) mice (Figure 1f). This is consistent with the fact that CD16 on the granulocytes from the patients are more sensitive than GPI-APs of fibroblasts. Western blot analysis of the brain lysates from *Pigo*^{b/-}(KI/KO) mice showed slight but significant decrease in GPI-APs and MBP (Supplementary Fig. 4).

- *Moreover, it is not shown which is the effect of these mutations on *Pigo* expression at mRNA and protein level. This is a crucial analysis to understand, for instance, why the T130N mutation (*Pigob/b*) is associated with a more severe pathological phenotype in comparison with the other two mutations.*

Moreover, as stated by the authors in the discussion [In human cells, Arg119Trp mutant showed the most severe decrease in activity leading to the severest clinical symptoms, such as intractable seizures, severe intellectual disability, and multiple organ anomalies, whereas, in the murine cells, the severest case was the Thr130Asn mutant], there is a difference between mouse and human cells regarding the effect of these mutations. Thus, for example, a more detailed characterization through RT-PCRs and Western blot analyses is needed to solve this discrepancy.

This is a very important point and thank the reviewer for raising it up. As shown in the previous paper of *PIGO* deficiencies, protein expressions of Arg119Trp and Thr130Asn mutants were not decreased (Tanigawa, J., *et al. Hum Mutat* **38**, 805-815 ,2017). In the revised manuscript, we added the WB analysis data of *Pigo*-KO Neuro2a cells transfected with the mutant *Pigo* cDNAs (Supplementary Fig. 1). Mouse *Pigo* mutant proteins were not decreased compared to the wild type *Pigo*. In fact, according to the degrees of the restoration of the GPI-APs on the *PIGO*- or *Pigo*-KO cells rescued by human or mouse mutant cDNAs, residual activities of both mutants (Arg119Trp and Thr130Asn) showed different order between human and mouse but with only slight difference. *PIGO* is associated with *PIGF*, which is critical for *PIGO* activity. One possible explanation is that these mutations affect the conformation of the *PIGO*/*PIGF* complex differently in human and mouse. It is true that decreased residual activities correlate the clinical severity, but in case of human, genetic background is different each other, clinical severity varies even among the patients with the same mutations. For this reason, Arg119Trp mutation might not always causes severer symptoms in human.

- *The second part remains more problematic and less than conclusive. Based on the presented data, it appears that the HITI-mediated integration of WT Pigo cDNA upstream the endogenous mutant gene poorly contributes in ameliorating the pathological phenotype and probably is responsible for the aberrant mRNA products observed in the liver. Indeed, as shown in Fig. 6b only a minimal proportion of the corrected Pigo transcripts results from the donor integration (< 3%), with a more preponderant contribution from the ITR-mediated AAV transcription. At this point, the exact contribution of the donor integration in the phenotypic rescue needs to be stringently ascertained. For this, authors need to test both in Pigo-KO Neuro2a cells and in Pigo^{b/b} mice the overall effects (gene expression and mouse behavior) of the donor DNA (pAAV mPigoHITIgRNA) without the Cas9. This is the most rigorous way to ascertain whether the Cas9-dependent HITI-mediated integration significantly contributes or not to the pathological phenotypic rescue.*

Thank you for picking up the important points. In this revised version, per the reviewer's suggestions, we compared the efficacies of HITI-TE and single administration of AAV- donor, which allows ITR-dependent expression of *Pigo*, in in vitro and in vivo experiments. (Fig. 7).

In in vitro experiments, *Pigo*-KO Neuro2a cells were transfected with pAAVCas9 and pAAV donor (pAAV-mPigoHITIGRNA) or pAAV donor only, and analyzed by FACS to determine GPI-AP- restored population (Fig. 7a). The rescue effect of AAV donor only clearly seen on day 3 and rapidly disappeared within 2 weeks most likely due to the dilution of pAAV donor plasmid by cell proliferation. In contrast, the cells transfected with both pAAVCas9 and pAAV donor stably retained the restored population (20% of total) for 18 days. This result strongly suggests the advantage of HITI-mediated integration over ITR-mediated ectopic expression for dividing cells in terms of duration of rescued GPI-AP expression. Similar tendency was observed in vivo situation. To clarify the contribution of AAV donor only in mice, *Pigo*^{b/-} mice were treated with AAVCas9 and AAV donor (i.e. HITI-TE treated) or treated AAV donor only. In the first 8 to 10 weeks, there were similar effects in both groups, however, the body weight of mice treated with AAV donor only tended to decline from 10 ~12 weeks of after the treatment (Fig. 7b). The elevation of Gr-1 (a GPI-AP) levels on blood granulocytes and amelioration of hyperphosphatasia were shown in *Pigo*^{b/-} mice after HITI treatment, which is clearly better than AAV donor only treatment (Fig 7c and 7d). The differences between two groups were not statistically significant probably due to the limited numbers of the mice. These results indicate that both HITI-mediated integration and ITR-driven transient expression of *Pigo* from the AAV donor contributed to the therapeutic effects with HITI-TE treatment.

- *Another troublesome aspect of this study is related to the only use of ITRs to express the therapeutic gene. The promoter activity of the ITR is poorly studied and its cell-type specificity and temporal dynamics is unknown. Moreover, authors did not compare Pigo transgenic expression triggered by the ITRs with a wild-type control. Thus, it has not been assessed if AAV-mediated Pigo expression resembles its physiological expression in time and space. Detailed analyses of ITR-dependent activity in different cell types and tissues and its temporal evolution need to be described in detail to support its potential therapeutic validity.*

We agree with the reviewer that the promoter activity of the ITR is poorly studied. In this revision, we compared promoter activities of AAV ITR and the endogenous *Pigo*. NIH3T3 cells were infected at various MOI with AAV-PHP.eB bearing nEF promoter-driven EGFP to measure the infection efficiencies of AAV-PHP.eB by flow cytometry (Supplementary Fig. 10a and 10b). In parallel, NIH3T3 cells were infected at various MOI with donor AAV and RT-qPCR were performed for the *Pigo* expression (Supplementary Fig. 10c). Viral genome copy numbers in the infected cells were also determined by qPCR (center panel in Supplementary Fig. 10d). Calculating from the viral genome copy numbers and *Pigo* mRNA levels, normalized by the infection efficiencies by FACS, promoter activity of ITR was 3~12 times of endogenous promoter activity in NIH3T3 cells. Thus, the ITR promoter activity was comparable to endogenous *Pigo* promoter in NIH3T3 cells, however detailed analysis of ITR-dependent activity in different tissues and its temporal regulation need to be investigated for its therapeutic use.

- Minor revision:

1) In Figure 1 a Sanger sequence interferogram is required to show the presence of the mutations

In this revision, we performed Sanger sequences and showed the mutations (Supplemental Fig. 3).

- *2) In Figure 3 the efficiency (on-target analysis) and the selectivity (off-target analysis) of the selected sgRNA should be included*

Per reviewer's request, we have now provided 5'RACE data in Neuro2a transfected with pAAVCas9 and pAAV donor and added the results in Figure 6 not in Figure 3 (right panel in Fig. 6b). Clear off-target effect was not observed in the Neuro2a cells. Furthermore, we provide FACS analysis of the restored expression of CD24 (GPI-AP), PCR and Sanger sequencing data which demonstrate the efficacy and accuracy of on-target effects (Fig. 3d-f).

- *3) In Figure 5 the 5' junction PCR in brain should be included*

We have now added the data in Fig. 5b and c.

- *4) In Extended Figure 3 a graph representing the Western blot quantification should be included*

We have analyzed 3 mice and showed in the graph of Supplementary Fig. 4.

- *5) Behavioral tests are presented only for the A-line KI mice while it should be presented also for the other 2 lines.*

B-line mice were very small. They showed neurological dysfunctions such as ataxic gate and tremor. In these situations, it is difficult to validate the results of behavioral test. Instead, we performed the hanging test for evaluating the muscle strength before and after the treatment. Symptoms of C-line mice were milder than A-line mice. According to the results from A-line mice, it seemed that there would be no significant difference compared to the wild type mice by usual behavioral tests. It would need the specific behavioral tests focusing on the higher brain function, which we could not perform in this paper because of limited time and limited numbers of C-line mice.

We added the sentences in the result why we selected A-line mice for the behavioral test.

- Reviewer #2 (Remarks to the Author):

This is an impressive study aimed at developing and treating a mouse model of the human genetic disorder, PIGO deficiency. The authors accomplish this and provide important insights into 1.) identifying the best mutation to analyze and 2.) using cleverly designed integrating and extra chromosomal AAV-based vectors of the normal allele. This well written presentation offers an approach which could be generalized to study and treat other rare diseases where multiple mutations in a single gene present challenges of both phenotype and rescue strategy.

Previously described knockout PIGO animals are nonviable and any useful model would require knockin the corresponding mutations. The authors create and evaluate the impact of 3 knockin mutations (a, b, and c) using cell based assays and then create 3 mouse lines carrying each homozygous mutation or as a heterozygote with the null allele, e.g., b/- (T130N/-). Each of these lines was extensively evaluated for growth, neurological, structural, behavioral, and biochemical phenotypes. There was a graded response depending on the mutations and whether they were homozygous b/b or b/-. The extensive phenotyping provides a solid genotype-phenotype correlation. Only the severest mutation (b/- and b/b) recapitulated patient PIGO phenotype as well demonstrated in the videos of b/b mice.

Treatment by gene replacement therapy using rAAV is possible, but the need to treat very young patients means that the extra chromosomal normal allele would be diluted in time. Furthermore, PIGO forms a complex with another protein (PIGF) which also interacts with PIGG. Unregulated over expression of PIGO could upset the appropriate distribution of PIGF. Ideally, integrating the rescuing gene under the control of its endogenous promoter would be preferred. To this end, the authors applied their previously published non-homologous end joining (NHEJ)-based homology-independent strategy for targeted transgene integration (HITI) giving higher integration into dividing and non-dividing cells. They combined this with an extra chromosomal weak promoter of the AAV vector to keep expression to a modest level.

The single intravenous dose 100 billion genome copies of the therapeutic vectors into newborn of b/- and b/b mice had significant beneficial effects in both strains in terms of increased appearance of GPI-anchored protein (Gr-1) on granulocytes, decrease of plasma alkaline phosphatase indicating return of functional GPI synthesis. Low body weight and slow growth also improved. Tremors, grip strength and hindlimb claspings, and seizures also improved with the therapy. The extrachromosomal AAV appeared to account for most (60-80%) of the transcripts in the brain and only 3-5% from the integrated version. At this point the off-target effects appeared modest

- *Overall this is a novel and compelling paper about the therapeutic potential of a gene therapy for PIGO deficient patients. While more work is needed before this is ready for human trials, it*

provides a solid foundation and an exciting step forward.

We would like to thank the reviewer for such an encouraging comments.

- *My only suggestion is to complement the extensive genotype phenotype studies with a biochemical assay of PIGO activity. While this is not essential, it would provide a perspective of phenotype and residual activity. As more PIGO mutations are identified, functional assay data might provide a database of likely disease course. Perhaps a simple TLC plate analysis could do this.*

We appreciate the reviewer's suggestions. For the functional analysis, we believe that the most sensitive method to validate the mutant PIGO activity is measuring GPI-AP rescued level using FACS analysis of PIGO knockout cells transfected with mutant PIGO cDNA compared with wild type PIGO cDNA. We provided this data in Supplemental Figure 1 and also in reference 22.

We added the explanation for Supplementary Fig. 5b because it was missing in the previous version.

Reviewers' Comments:

Reviewer #1:

Remarks to the Author:

Kuwayama and colleagues have performed several new experiments improving the quality of the manuscript. Despite this, some of my previous concerns have not been satisfactorily answered. Regarding the first part (generation and characterization of KI and KO mouse models for Pigo gene), I have appreciated the inclusion of histological and cellular analyses to characterize Pigo^{b/-} mutant mice (supplementary figure 7). I suggest to the authors to include high quality images and quantification to enhance the quality of these results. In addition, it is not completely clear to me why authors performed the *in vivo* characterization in Pigo^{b/-} mutant mice, whereas *in vitro*, they studied neurons derived from Pigo^{b/b} fetus (supplementary figure 8). Can they elucidate this choice?

Nevertheless, my major concerns refer to the second part of the manuscript (optimization of a novel gene therapy approach to restore wild-type Pigo expression). In fact, despite I appreciate their work, the authors compared the effect of the donor alone (TE) against the combination with Cas9 (HITI-TE) without performing behavioral analyses (Figure 7). Thus, I think that the same behavioral analyses presented in figure 4f and 4g should be included to consolidate the role of HITI in the therapeutic effect.

Then, the authors studied ITR-promoter activity showing a comparison with the endogenous Pigo promoter. Nevertheless, this analysis was performed exclusively *in vitro* in NIH3T3 cells (supplementary figure 10), thus, in my opinion, the result cannot be considered predictive of the *in vivo* experiments. In fact, the AAV ITR has never been exploited as a promoter for a gene therapy approach, so a critical characterization in organs is crucial to establish its therapeutic potential. I understand that a full analysis of this kind might require extensive time, but at least a characterization of AAV copy number and transgene expression at RNA levels in the mouse brain should be included.

In summary, the authors should analyze through molecular assays the effect of the donor alone in the brain of Pigo mutant mice, in order to characterize the activity of the ITR as a promoter. Then, they need to compare at behavioral level mice treated with the donor with or without the Cas9 to determine the contribution of HITI on the therapeutic effect.

In overall, despite the amount of work done by authors, some of the answers to my previous points need to be carefully revised. In particular, for reasons discussed above, new experiments are required to consolidate the potential of HITI-TE as a new therapeutic approach.

Reviewer #2:

Remarks to the Author:

Since this is a revision, I have already provided my opinion about the importance of this work, its methods and conclusions. So no need to repeat.

Response to the reviewers

Reviewer #1 (Remarks to the Author):

Kuwayama and colleagues have performed several new experiments improving the quality of the manuscript. Despite this, some of my previous concerns have not been satisfactory answered.

Regarding the first part (generation and characterization of KI and KO mouse models for *Pigo* gene), I have appreciated the inclusion of histological and cellular analyses to characterize *Pigob*⁻ mutant mice (supplementary figure 7). I suggest to the authors to include high quality images and quantification to enhance the quality of these results.

- ✓ According to the reviewer's suggestion, we exchanged figures (x60 images of O4, GFAP, Contactin2 and Synapsin) and showed the quantification (supplementary figure 7).

In addition, it is not completely clear to me why authors performed the *in vivo* characterization in *Pigob*⁻ mutant mice, whereas *in vitro*, they studied neurons derived from *Pigob*^b fetus (supplementary figure 8). Can they elucidate this choice?

- ✓ These gene manipulated mice are not highly reproductive for unknown reasons and availability of embryos/pups are limited. Although we tried to use mice of the same genotype in similar tests, this was not met in some cases. Although phenotype of *Pigo* ^b/⁻ mutant mice was a bit severer than that of *Pigo* ^b/^b mutant mice, their phenotypes were mostly similar.

Nevertheless, my major concerns refer to the second part of the manuscript (optimization of a novel gene therapy approach to restore wild-type *Pigo* expression). In fact, despite I appreciate their work, the authors compared the effect of the donor alone (TE) against the combination with Cas9 (HITI-TE) without performing behavioral analyses (Figure 7). Thus, I think that the same behavioral analyses presented in figure 4f and 4g should be included to consolidate the role of HITI in the therapeutic effect.

- ✓ Now we added data with the hanging tests (Fig. 7e) and data about tremor (Fig. 7f). We made the AAV treated mice with *Pigo* ^b/^b mutant mice because we had the data

of hanging tests comparing HITI treated mice with non-treated mice (Fig. 4g). We added the following sentences in the manuscript.

“AAV donor only treated *Pigo*^{b/b} mice showed clearly improved performance in the hanging test with both 5 mm and 11 mm meshes (Fig.7e) compared with the untreated *Pigo*^{b/b} mice (data in Fig. 4g). In both tests, HITI-TE treated mice had slightly longer mean hanging time than AAV donor only treated mice (139 sec vs 126 sec with 5mm mesh and 180 sec vs 134 sec with 11mm mesh) (Fig. 7e). Four out of six mice treated with donor only hung unstably during the tests, whereas all the HITI-TE treated mice hung stably and moved around. HITI-TE treatment ameliorated tremor better than AAV donor only treatment in both timing of tremor appearance and the severity scores (Fig. 7f and Fig. 4f). These results indicate that both HITI-mediated integration and ITR-driven transient expression of *Pigo* from the AAV donor contributed to the therapeutic effects with HITI-TE treatment.”

Then, the authors studied ITR-promoter activity showing a comparison with the endogenous *Pigo* promoter. Nevertheless, this analysis was performed exclusively *in vitro* in NIH3T3 cells (supplementary figure 10), thus, in my opinion, the result cannot be considered predictive of the *in vivo* experiments. In fact, the AAV ITR has never been exploited as a promoter for a gene therapy approach, so a critical characterization *in organs* is crucial to establish its therapeutic potential. I understand that a full analysis of this kind might require extensive time, but at least a characterization of AAV copy number and transgene expression at RNA levels in the mouse brain should be included.

- ✓ Thank you for the important suggestion. We analyzed the ITR driven expression compared to the *Pigo* endogenous expression in the mouse brain. ITR promoter activity was similar to endogenous *Pigo* promoter activity in the cerebrum. The data is now shown in Supplementary Fig. 11. We also modified the explanation of *in vitro* study with NIH3T3 cells in the manuscript.

In summary, the authors should analyze through molecular assays the effect of the donor alone in the brain of *Pigo* mutant mice, in order to characterize the activity of the ITR as a promoter. Then, they need to compare at behavioral level mice treated with the donor with or without the Cas9 to determine the contribution of HITI on the therapeutic effect.

In overall, despite the amount of work done by authors, some of the answers to my previous points need to be carefully revised. In particular, for reasons discussed above,

new experiments are required to consolidate the potential of HITI-TE as a new therapeutic approach.

- ✓ We now have sets of comparing data with HITI-TE treatment and AAV donor only treatment about body weight, GPI-AP levels, serum ALP, hanging tests and tremor observation. These data convincingly suggest that *Pigo* expression mediated by both HITI-mediated target integration and AAV ITR promoter activity contribute to the therapeutic effects of HITI-TE.

Reviewers' Comments:

Reviewer #1:

Remarks to the Author:

Although the authors revised their study a couple of times, unfortunately the overall design and conclusions of this work remain not sufficiently strong and convincing in several parts and for multiple reasons. First and above, the authors' claiming of the relevance of the combination of the dual treatment with HITI and ITR-base transgene expression is substantially not justified from the presented results. In fact, the efficiency of the HITI-based gene repair is less than 3% in the whole brain tissue. It is difficult to believe that this fraction might have a substantial impact in the rescue of the severe defects and diffuse alterations caused by this pathology without direct evidence. The authors tried to justify this suggesting a non-cell autonomous effect, but this remains only speculative since no results directly or indirectly are presented supporting this hypothesis. Most importantly, it is not presented the rescue with only the HITI procedure to confirm that such a low grade of gene repair can sustain such an important rescue. Moreover, when the authors compared the rescue with the combined treatment or with only the AAV donor vector, the rescue in weight, Gr-1 levels, latency to fall were not significantly different between the two treatments. Thus, the HITI-base gene repair seems to have a rather negligible effect.

In addition, it is lacking a reasonable rational for the expression of the transgene by the viral ITR. In fact, the same combination therapy could be designed including a promoter upstream to the sgRNA sites in the viral backbone to sustain higher level of the transgene while enabling HITI-mediated gene repair anyway. Thus, the use of the viral ITR is a poor choice in order to design an efficient gene therapy strategy. Other aspects remain still confusing as their analysis of gene expression presented in Figure S11 where it is not clear how they calculated the transgene expression levels presenting extremely high copy number (higher than 8000) in AAV-treated and control animals.

In overall, this gene therapy strategy has substantial flaws that the authors seem not to be aware, and their conclusions are not supported by the results described in their study.

Reply to the Reviewer #1

Thank you very much for taking time to re-review our paper and giving us the constructive comments. The following are the point-by-point responses to the comments.

Although the authors revised their study a couple of times, unfortunately the overall design and conclusions of this work remain not sufficiently strong and convincing in several parts and for multiple reasons. First and above, the authors' claiming of the relevance of the combination of the dual treatment with HITI and ITR-base transgene expression is substantially not justified from the presented results. In fact, the efficiency of the HITI-based gene repair is less than 3% in the whole brain tissue. It is difficult to believe that this fraction might have a substantial impact in the rescue of the severe defects and diffuse alterations caused by this pathology without direct evidence. The authors tried to justify this suggesting a non-cell autonomous effect, but this remains only speculative since no results directly or indirectly are presented supporting this hypothesis.

As the reviewer says, it is surprising that as little as 10% correction (by the combination of HITI mediated gene editing and ITR driven transgene) could rescue the mouse phenotype drastically. However, as we already described in the discussion, there are several reports showing less efficient correction could improve the phenotype of disease model mouse. As reported previously, gene-correction of only 4% in retina by HITI was sufficient to alleviate some of the phenotypes associated with retinitis pigmentosa in rats (Suzuki K, et al., *Nature* 2016). Furthermore, gene correction of 2% in liver in a premature aging mouse model by SATI (another gene editing method) could diminish aging phenotypes in several tissues as well as an extension of lifespan (Suzuki K, et al., *Cell Research* 2019). Another group showed that cationic lipid-mediated *in vivo* delivery of Cas9:guide RNA complexes can ameliorate hearing loss in a mouse model of human genetic deafness. They only showed *in vitro* efficiency using the fibroblast (10%) (Gao X et al., *Nature*. 2018). The other group reported that 0.4% of gene correction in hepatocytes rescued the body weight loss phenotype in a mouse model of the human disease hereditary tyrosinemia (Yin, H. et al., *Nat Biotechnol.* 2014). Among these, as for the last one, improvement was achieved by expansion of the gene-corrected cells. However, for the other cases, we don't have any clear explanation. This might be accounted for by an indirect non-cell-autonomous effect from the small

number of gene-corrected cells. There are several evidence published showing that GPI anchored proteins (GPI-APs) are transferred to the other cells. GPI-APs expressed on the surface of transgenic mouse red blood cells were transferred in a functional form to endothelial cells in vivo (Kooyman DL, et al. *Science* 1995)". *Piga*⁻ embryoid bodies (EB) produced no secondary hematopoietic colonies, however, in the chimeric EB composed with normal cells and *Piga*⁻ cells, hematopoiesis from knock-out cells was reconstituted due to transfer of GPI-APs from normal to knock-out cells (Dunn, DE. et al., *PNAS* 1996). GPI-AP are efficiently sorted in exosomes and are transferred to the other cells (Müller, GA. et al. *Biochim. Biophys. Acta* 2018; Vidal, M. et al., *Adv. Drug Deliv. Rev.* 2020) According to these lines of evidence, GPI-APs transfer from the gene-corrected cells might contribute to the phenotype improvement.

We added following sentences in the Discussion.

*"There are several lines of evidence published showing that GPI anchored proteins (GPI-APs) are transferred to the other cells. GPI-APs expressed on the surface of transgenic mouse red blood cells were transferred in a functional form to endothelial cells in vivo (Kooyman DL, et al. Science 1995)". *Piga*⁻ embryoid bodies (EB) produced no secondary hematopoietic colonies, however, in the chimeric EB composed with normal cells and *Piga*⁻ cells, hematopoiesis from knock-out cells was reconstituted due to transfer of GPI-APs from normal to knock-out cells (Dunn, DE. et al., PNAS 1996). GPI-APs are efficiently sorted in exosomes and are transferred to the other cells (Müller, GA. et al. Biochim. Biophys. Acta 2018; Vidal, M. et al., Adv. Drug Deliv. Rev. 2020) According to these lines of evidence, GPI-APs transfer from the gene-corrected cells might contribute to the phenotype improvement."*

Most importantly, it is not presented the rescue with only the HITI procedure to confirm that such a low grade of gene repair can sustain such an important rescue.

In case of our disease model, it was impossible to show the rescue with only the HITI procedure. As the missense mutation was located in the first exon, we had to construct the AAV donor vector containing the full length of *Pigo* cDNA to be integrated into the 5' UTR of *Pigo* gene. We hypothesized that AAV donor containing the full length cDNA works not only as a donor of HITI mediated integration but also as a transgene, which can contribute to the expression of the normal protein.

Previously, the pure effect of HITI in the other disease model, retinitis pigmentosa in rats was reported (Suzuki K, et al., *Nature* 2016). In that case, the deletion of the splicing donor site of exon2 caused the disease, and the constructed AAV donor vector contains only a normal exon2 and inserted into intron1 by HITI. Similar to our disease model, gene-correction of only 4% in retina by HITI was sufficient to alleviate some of the phenotypes associated with retinitis pigmentosa in rats.

We added the following sentences in the Discussion.

“In case of our disease model, as the missense mutation was located in the first exon, we had to take the strategy of the target insertion of full-coding Pigo cDNA into the 5'UTR of the Pigo gene. It should be safe even if it causes indels at the target site and can achieve an endogenous expression level because the inserted Pigo cDNA will be regulated by the endogenous promoter and the regulatory system. Therapeutic effect was not solely by HITI mediated integration but also by the extra-chromosomal AAV donor containing full Pigo cDNA”

Moreover, when the authors compared the rescue with the combined treatment or with only the AAV donor vector, the rescue in weight, Gr-1 levels, latency to fall were not significantly different between the two treatments. Thus, the HITI-base gene repair seems to have a rather negligible effect.

Although statistical significance was not always obtained because of a large variation of the effect among the individual mice, the average scores were better in all parameters with HITI treated mice, especially in Gr-1 levels, achieving significantly better improvement than the Donor only group. It should also be noted that because some fraction of the Donor vector might be inactivated by Cas9 plus guide RNA in the combination therapy, contribution of the Donor-derived mRNA should be smaller than in the Donor only therapy, further supporting a contribution of HITI to the efficacy of the combination therapy.

In addition, it is lacking a reasonable rationale for the expression of the transgene by the viral ITR. In fact, the same combination therapy could be designed including a promoter upstream to the sgRNA sites in the viral backbone to sustain higher level of the

transgene while enabling HITI-mediated gene repair anyway. Thus, the use of the viral ITR is a poor choice in order to design an efficient gene therapy strategy.

The referee is right in that use of exogenous promoter is possible in the combination therapy. We did not choose that way because we wanted to have a low *Pigo* expression in order not to affect balance between *Pigo* and *Pigg*. *Pigo* and *Pigg* are individually complexed with *Pigf* and mediate EthN-P transfers to *Man3* and *Man2*, respectively. It was shown previously that PIGO over-expression affected PIGG enzyme level by interfering generation of PIGG and PIGF complex (Shishioh, N. et al. J Biol Chem. 2005). Recently, it became clear that too much expression of transgene is harmful in gene therapy, causing liver dysfunction and thrombocytopenia. Clinicians are trying to regulate the expression using drug inducible constructs. As ITR driven expression of *Pigo* is at a similar level to endogenous expression, it would be a reasonable choice in combination with gene editing therapy.

We added the following sentences in the Discussion

“The same combination therapy could be designed including a promoter upstream to the sgRNA sites to sustain higher level of the transgene while enabling HITI-mediated gene repair, however we wanted to have a low Pigo expression in order not to affect balance between Pigo and Pigg. Pigo and Pigg are individually complexed with Pigf and mediate EthN-P transfers to Man3 and Man2, respectively. It was shown previously that PIGO over-expression affected PIGG enzyme level by interfering generation of PIGG and PIGF complex³¹. Recently, it became clear that too much expression of transgene is harmful in gene therapy, causing liver dysfunction and thrombocytopenia. Clinicians are trying to regulate the expression using drug inducible constructs. As ITR driven expression of Pigo is at a similar level to endogenous expression, it would be a reasonable choice in combination with gene editing therapy.”

Other aspects remain still confusing as their analysis of gene expression presented in Figure S11 where it is not clear how they calculated the transgene expression levels presenting extremely high copy number (higher than 8000) in AAV-treated and control animals.

As for the method for the copy number calculation in vivo, we used lysate of whole cerebrum and calculated based on the standard curve of absolute copy number made by the serial dilution of expression plasmid. Based on the calculation from the genomic

gene copy number (approximately 30,000), the lysate used in Figure S11c corresponded to approximately 15,000 cells. therefore 8,000 copy- expression of *Pigo* cDNA does not seem so high. We changed the label on the vertical axis of the RNA expression from relative copy number to relative expression. RNA and genomic DNA were isolated simultaneously from the lysate using Allprep DNA/RNA mini kit and the same aliquot of DNA and RNA were used for analysis of ITR driven and endogenous *Pigo* expression. Thanks to your suggestion in the second revision, we by performing *in vivo* analysis obtained this important result that the activity of ITR promoter is at a similar level to that of endogenous *Pigo* promoter as shown in Figure S11c and e.

We add the scheme of the experimental procedure in Figure S11a.